# Charge-transfer regulated visible light driven photocatalytic $H_2$ production and $CO_2$ reduction in tetrathiafulvalene based coordination polymer gel

Parul Verma[1], Ashish Singh[1], Faruk Ahamed Rahimi[1], Pallavi Sarkar[2], Sukhendu Nath [3], Swapan Kumar Pati[2] & Tapas Kumar Maji [1✉]

The much-needed renewable alternatives to fossil fuel can be achieved efficiently and sustainably by converting solar energy to fuels via hydrogen generation from water or $CO_2$ reduction. Herein, a soft processable metal-organic hybrid material is developed and studied for photocatalytic activity towards $H_2$ production and $CO_2$ reduction to CO and $CH_4$ under visible light as well as direct sunlight irradiation. A tetrapodal low molecular weight gelator (LMWG) is synthesized by integrating tetrathiafulvalene (TTF) and terpyridine (TPY) derivatives through amide linkages and results in TPY-TTF LMWG. The TPY-TTF LMWG acts as a linker, and self-assembly of this gelator molecules with $Zn^{II}$ ions results in a coordination polymer gel (CPG); Zn-TPY-TTF. The Zn-TPY-TTF CPG shows high photocatalytic activity towards $H_2$ production (530 µmol $g^{-1}h^{-1}$) and $CO_2$ reduction to CO (438 µmol $g^{-1}h^{-1}$, selectivity > 99%) regulated by charge-transfer interactions. Furthermore, in situ stabilization of Pt nanoparticles on CPG (Pt@Zn-TPY-TTF) enhances $H_2$ evolution (14727 µmol $g^{-1}h^{-1}$). Importantly, Pt@Zn-TPY-TTF CPG produces $CH_4$ (292 µmol $g^{-1}h^{-1}$, selectivity > 97%) as $CO_2$ reduction product instead of CO. The real-time $CO_2$ reduction reaction is monitored by in situ DRIFT study, and the plausible mechanism is derived computationally.

[1] Molecular Materials Laboratory, Chemistry and Physics of Materials Unit, School of Advanced Materials (SAMat), Jawaharlal Nehru Centre for Advanced Scientific Research, Jakkur, Bangalore 560 064, India. [2] Theoretical Sciences Unit, School of Advanced Materials (SAMat), Jawaharlal Nehru Centre for Advanced Scientific Research, Jakkur, Bangalore 560 064, India. [3] Ultrafast Spectroscopy Section, Radiation & Photochemistry Division, Bhabha Atomic Research Centre, Mumbai 400 085, India. ✉email: tmaji@jncasr.ac.in

Artificial photosynthesis, i.e. the conversion of sunlight into fuels, is a green approach and has the potential to solve the global energy crisis. In recent years, a significant amount of research has been carried out to develop artificial systems[1] for mimicking the sophisticated methodology of nature's water splitting[2,3] as well as $CO_2$ reduction[4,5]. Natural photosynthesis rely on the occurrence of precise sequences of proteins/enzymes to the several elementary steps of inter-component light-absorption, charge separation, and migration[6,7]. The synthetic assimilation of these parameters precisely in a spatial organization of molecular components is indeed a challenging task[8,9]. The recent upsurge of converting $CO_2$ into fuels like $CH_3OH$, $CH_4$, or different chemical feedstock has gained widespread attention[10–12]. The conversion of $CO_2$ to hydrocarbon fuels would mitigate not only the effect of $CO_2$ concentration in the atmosphere but also reduce the dependency on fossil fuel-based economy. However, the photoreduction of $CO_2$ molecules is a complex and challenging process due to the very high dissociation energy of the C=O bond (~750 kJ/mol)[13]. Only a handful of metal[14,15], metal oxide[16–18], and chalcogenides[19,20] based heterogeneous catalysts were reported for photocatalytic $CO_2$ reduction to $CH_4$, but most of them suffer from a low conversion efficiency and poor selectivity[21,22]. $CH_4$ formation is thermodynamically favourable ($E^0 = -0.24$ V versus RHE at pH = 7) than CO formation ($E^0 = -0.53$ V versus RHE at pH = 7)[23,24] as the former reaction takes place at a lower potential. Nevertheless, from a kinetic point of view, the eight-electron reduction of $CO_2$ to $CH_4$ is more difficult, especially under photochemical condition than the two-electron reduction of $CO_2$ to CO[25]. To address challenges associated with photochemical $H_2$ production and $CO_2$ reduction, a novel photocatalytic system needs to be developed by the innovative design of photosensitizer and catalytic moiety[26,27]. Recently, carbon-nitride based photocatalyst for $H_2$ evolution and $CO_2$ reduction to CO has been reported[28,29]. Moreover, there is a huge lacuna in designing and developing such versatile photocatalyst materials that can reduce both, water and $CO_2$ efficiently.

To this end, developing soft hybrid materials, such as coordination polymer gel (CPG), assembled by the low molecular weight gelator (LMWG) based linker and suitable metal ions, could be an excellent design approach in the realm of photocatalysis[30,31]. Such hierarchical soft nanofibrous materials[1,32,33] can facilitate the facile diffusion of reactants to the active sites and will show efficient electron transfer between different components[34–36]. These artificial hybrid synthetic systems can mimic the intricate functioning of the natural photosystem and can eventually show impressive $H_2$ evolution from water[37–40] or $CO_2$ reduction. Extended face-to-face arrays of the donor–acceptor[38] π-chromophoric systems would be an ideal candidate for light harvesting[41]. These systems will allow greater exciton mobility, which, in turn, leads to charge generation and subsequent electron transfer to the catalyst[36,40]. To this end, tetrathiafulvalene (TTF) moiety is a well-known p-type[42] semiconductor possessing high electron donation capability with excellent photostability and good charge carrier mobility. Further, the integration of a suitable electron acceptor unit to the TTF moiety could result in a system with excellent charge-transfer characteristics[43]. Thus, designing a TTF containing donor–acceptor[44,45] based LMWG could be an elegant approach for developing new photocatalyst material. Such systems are likely to show low energy charge transfer in the visible range, which will further reduce the bandgap, and corresponding visible-light photocatalyst can be realized[45]. Coordination driven array of donor–acceptor pairs would further improve photocatalytic performances by enhancing charge transfer to the catalytic centre. Thus, the introduction of a suitable metal-binding moiety like terpyridine on TTF containing LMWG could lead to the formation of CPG and provide an opportunity for further improving the photocatalytic activity[46].

Here, we aim to report an emerging class of materials known as 'coordination polymer gel' by integrating $Zn^{II}$ with TTF-based LMWG and explored as a photocatalyst for solar fuel production based on water and $CO_2$ reduction. The intermolecular charge transfer is regulated by the innovative design of LMWG, where TTF derivative is connected with metal-binding terpyridine units (TPY) through a flexible alkyl amide chain. The coordination polymer gel (Zn-TPY-TTF CPG) provides a suitable platform for light harvesting as well as a catalytic center for $H_2$ production from water (530 μmol $g^{-1}h^{-1}$) and $CO_2$ reduction to CO (438 μmol $g^{-1} h^{-1}$) with 99% selectivity (Fig. 1). Furthermore,

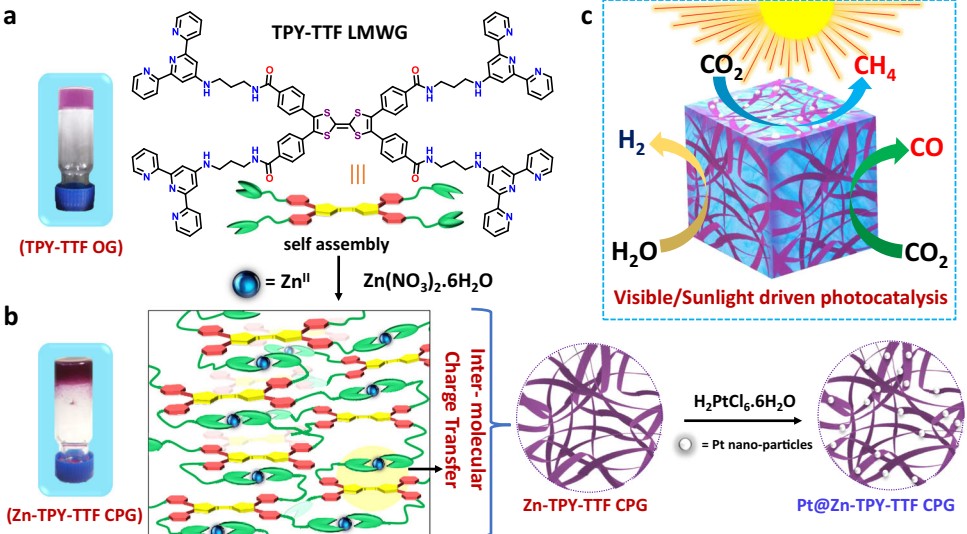

**Fig. 1 Schematic presentation of visible/sunlight-driven photocatalysis in TTF-based coordination polymer gel (CPG). a** TPY-TTF low molecular weight gelator (LMWG) based linker and corresponding organogel (TPY-TTF OG). **b** Self-assembly with $Zn^{II}$ toward the preparation of Zn-TPY-TTF CPG and preparation of Pt@Zn-TPY-TTF CPG by in situ stabilization of Pt nanoparticles on the Zn-TPY-TTF CPG. **c** Visible/sunlight-driven photocatalytic activity of Zn-TPY-TTF CPG and Pt@Zn-TPY-TTF CPG towards $H_2$ evolution and $CO_2$ reduction.

Zn-TPY-TTF CPG with nanoribbon morphology is conjugated with Pt co-catalyst, and the Pt@Zn-TPY-TTF CPG shows many folds enhanced photocatalytic activity towards $H_2$ production (14727 μmol g$^{-1}$h$^{-1}$). Interestingly, this Pt@Zn-TPY-TTF CPG produces $CH_4$ (292 μmol g$^{-1}$h$^{-1}$) as a photoproduct with high selectivity (97%) and impressive quantum efficiency. Importantly, both Zn-TPY-TTF CPG and Pt@Zn-TPY-TTF CPG shows the potential to perform sunlight-driven photocatalytic activity under ambient conditions. The in situ DRIFT study and DFT calculations help to elucidate the plausible mechanism of charge transfer regulated $CO_2$ reduction to CO/$CH_4$ formation using CPG catalysts under visible light as well as direct sunlight irradiation.

## Results

**Preparation and characterizations of TPY-TTF OG and Zn-TPY-TTF CPG.** The TPY-TTF LMWG was synthesized by the amide coupling reaction between 2,2′:6′,2″-terpyridin-4′-yl-propane-1,3-diamine (TPY-NH$_2$)[47] and 1,3,6,8-tetrakis (benzoic acid) tetrathiafulvalene (TTF(COOH)$_4$)[48] (details are provided in method section and supplementary information (SI)), Supplementary Figs. 1–2). The newly synthesized TPY-TTF LMWG was characterized by $^1$H and $^{13}$C Nuclear Magnetic Resonance (NMR), Fourier Transform Infrared (FT-IR) Spectroscopy, and Matrix Assisted Laser Desorption/Ionization -Time of Flight (MALDI-TOF) mass spectrometric analysis (Supplementary Figs. 3–6). UV-vis absorption study was performed for a well-characterized TPY-TTF LMWG in methanol (10$^{-6}$ M) and showed distinguished absorption bands at 270 nm and 320 nm corresponding to π→π* transition for TPY unit and TTF core, respectively (Fig. 2g). Notably, a low energy absorption band appeared at 520 nm that can be ascribed to intramolecular charge-transfer (CT) interaction between TTF core and benzo-amide moiety[48]. The CT property of TPY-TTF has also been supported by time dependant-density functional theoretical (TD-DFT) computation where the highest occupied molecular orbital (HOMO) and lowest unoccupied molecular orbital (LUMO) are centered in TTF and benzo-amide groups, respectively (Supplementary Fig. 7).

We have examined the gelation propensity of TPY-TTF LMWG in several solvent compositions (Supplementary Table 1). The purple-coloured opaque gel of TPY-TTF organogel (OG) was obtained in the solvent mixture (methanol (MeOH), dichloromethane (DCM) and water ($H_2O$) in 2:1:1 ratio) upon heating at 60 °C followed by cooling to room temperature as shown in Fig. 2a and characterization of gel was performed by different techniques (Supplementary Figs. 8–11). The strain-sweep rheology experiments for TPY-TTF OG at 25 °C showed the values of storage modulus (G′) and loss modulus (G″) move constantly in the linear viscoelastic (LVE) region, consisting of the larger G′ value under less strain range as compared to the G″, indicating the stable viscoelastic nature, which is a characteristic feature of a gel material (Supplementary Fig. 8). TPY-TTF OG was dried under vacuum at 80 °C to prepare the xerogel and studied the properties of the material (Supplementary Figs. 9–11). Morphology of the TPY-TTF OG xerogel was recorded by the Atomic Force Microscopy (AFM) and Field Emission Scanning Electron Microscopy (FE-SEM) that showed micron size staked layered type of morphology (Fig. 2b, c). We have also prepared aerogel using a critical point dryer (CPD) that also showed similar layered type morphology (Supplementary Fig. 11b). Transmission Electron Microscopy (TEM) images have further confirmed such morphologies (Fig. 2d). The distance between the layers in the stacked morphology was found to be 3.4 ± 0.4 nm based on AFM measurement. The high-resolution TEM analysis showed that the lattice fringes at ~3.7 Å (2d: inset) for the layered morphology

suggesting the self-assembly in TPY-TTF OG is driven by the intermolecular π–π interactions. This was also supported by the powder X-ray diffraction (PXRD) study of the xerogel that exhibited a peak at 2θ = 24° with a d-spacing of 3.7 Å (Fig. 2f). Further, we have performed DFT calculations which also support that the self-assembly is driven by intermolecular π–π stacking interactions between TTF---TPY units at a distance of 3.68 Å (Fig. 2e) and TTF---TTF units at a distance of 4.02 Å (Supplementary Fig. 12)[49,50]. The UV-vis absorption study for TPY-TTF OG xerogel displayed slightly red-shifted absorption as compared to the methanolic solution of TPY-TTF LWMG (Fig. 2g). It showed absorption bands at 300 nm and 330 nm, which can be assigned for π→π* transitions of TPY and TTF units of LMWG, respectively. Notably, a broad absorption between 480 and 615 nm was also observed, indicating the existence of charge transfer (CT) in TPY-TTF OG in the xerogel state. The closer analysis of the CT band showed that it consists of two distinguishable adjacent bands with absorption maxima at 510 and 565 nm (Fig. 2g; inset). Further, in-depth analysis through TD-DFT calculations showed CT absorption band at 498 nm and 564 nm, displaying quite fair agreement with the experimental results. The theoretical absorption at 498 nm was consist of both intramolecular (TTF to PhCONH-) and intermolecular (TTF to TPY) CT transitions (Fig. 4a, Supplementary Table 2). Whereas absorbance at 564 nm was attributed to the intramolecular (TTF-PhCONH-) CT transition. Based on the experimental and theoretical observations, the supramolecular assembly of TPY-TTF OG is represented in Fig. 2h, which showed that for both, TTF---TPY as well as TTF---TTF stackings are feasible. The optical bandgap for TPY-TTF OG, calculated by the Kubelka-Munk plot derived from UV-Vis diffuse reflectance spectrometry, was found to be 2.26 eV (Supplementary Fig. 13).

Next, the presence of four terpyridine units in the TPY-TTF gelator has prompted us to investigate further their metal-binding ability to develop coordination polymer gel (CPG) for widening their applications. To this end, we have chosen Zn$^{II}$ as a metal node for binding with TPY as such self-assembly is well explored due to soft acid-base interaction[46,51]. We have performed titration of TPY-TTF (8 × 10$^{-6}$ M in MeOH) with a methanolic solution of Zn(NO$_3$)$_2$.6H$_2$O (8 × 10$^{-4}$ M), and corresponding UV-vis absorption spectra were recorded (Fig. 3b). Notably, the presence of isosbestic point in the Zn$^{II}$ titration suggested the complex formation between Zn$^{II}$ and TPY-TTF LMWG (Supplementary Fig. 14a, b)[52]. The UV-vis titration study illustrated the binding of Zn(NO$_3$)$_2$ and TPY-TTF in the ratio of 2:1[47]. The association constant (K$_a$) of the Zn$^{II}$ ion with TPY-TTF was calculated to be 2.8 × 10$^4$ by the Benesi-Hildebrand plot (Supplementary Fig. 14c). Next, Zn(NO$_3$)$_2$ and TPY-TTF gelator was taken in a molar ratio of 2:1 in the solvent mixture of MeOH, DCM and $H_2O$ in 2:1:1 ratio, respectively. Heating the reaction mixture to 60 °C followed by cooling to room temperature has afforded a deep purple-coloured coordination polymer gel (Zn-TPY-TTF CPG) (Fig. 3a). Similar to OG, strain-sweep rheology experiments were performed for Zn-TPY-TTF CPG at 25 °C (Supplementary Fig. 15). The value of G′ and G″ upto 0.01% was found to be ~10 times larger for the Zn-TPY-TTF CPG than the OG, illustrating higher stability of the former one, which is possibly attained due to coordination of the Zn$^{II}$ ion[53]. More importantly, in both cases, tan δ (=G″/G′) value was found to be lower than one before reaching yield strain point, which is the intrinsic property of the gel phase (Supplementary Fig. 16). Next, the morphology of the Zn-TPY-TTF CPG was evaluated by FE-SEM upon drying the sample at 80 °C under vacuum, which showed the 3D entangled nanofibrous morphology (Fig. 3c). The aerogel of CPG was also prepared by the CPD method, which showed a

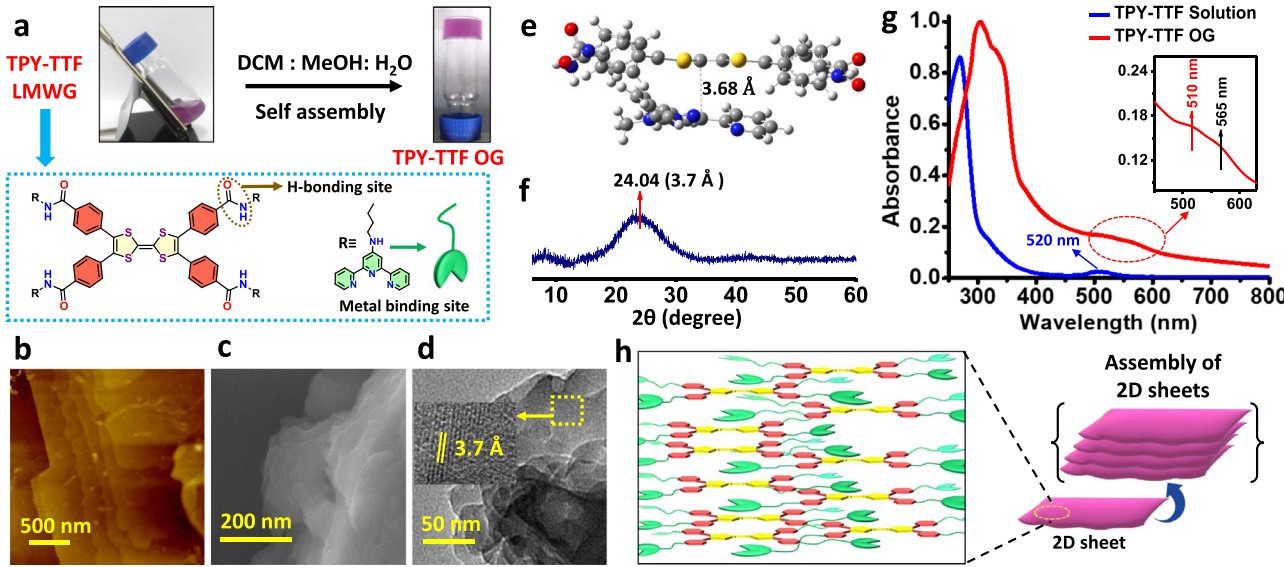

**Fig. 2 Preparation and characterizations of TPY-TTF OG. a** Photograph for organogel formation. Morphological analysis for xerogel. **b** AFM image. **c** FE-SEM image. **d** HR-TEM image (inset: showing lattice fringes). **e** Interplanar spacing in the optimized structure of TTF---TPY stacked model obtained through DFT calculation. **f** PXRD pattern for xerogel. **g** Comparison of absorption spectra of TPY-TTF LMWG in solution ($8 \times 10^{-6}$ M) with TPY-TTF OG xerogel state. **h** Pictorial representation for self-assembly in TPY-TTF OG forming layered sheet-like morphology.

**Fig. 3 Preparation and characterizations of Zn-TPY-TTF CPG. a** Photograph for Zn-TPY-TTF CPG formation. **b** UV-visible titration of TPY-TTF LMWG with $Zn^{II}$ ion. **c** FE-SEM image of Zn-TPY-TTF CPG xerogel. **d** HR-TEM image of Zn-TPY-TTF CPG xerogel. **e** Lattice fringes of the selected region. **f** AFM image of Zn-TPY-TTF CPG xerogel and corresponding, **g** height profile and **h** height histogram. **i** Comparison of absorption spectra of TPY-TTF OG and Zn-TPY-TTF CPG. **j** Interplanar spacing in the optimized structure of TTF---[Zn(TPY)$_2$]$^{2+}$ stacked model obtained through DFT calculation. **k** PXRD of Zn-TPY-TTF CPG xerogel. **l** Schematic representation for self-assembly of Zn-TPY-TTF CPG.

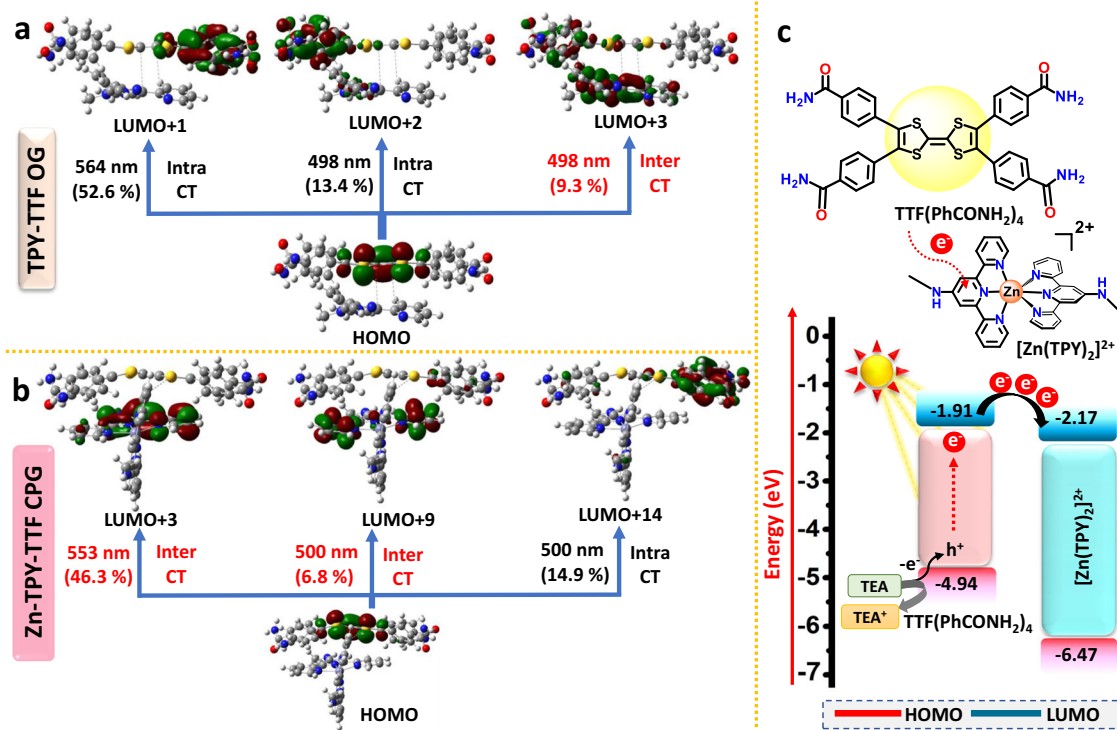

**Fig. 4 DFT calculations for CT interaction and band alignments in TPY-TTF OG and Zn-TPY-TTF CPG. a, b** HOMO-LUMO charge-transfer transitions with corresponding contributions from intramolecular and intermolecular charge transfer (CT) for TPY-TTF OG and Zn-TPY-TTF CPG, respectively. **c** HOMO-LUMO band alignments of TTF(PhCONH$_2$)$_4$ and [Zn(TPY)$_2$]$^{2+}$ for thermodynamic feasibility of electron transfer in the aqueous medium.

similar morphology suggesting that the fibrous morphology of Zn-TPY-TTF CPG is driven through metal coordination with LMWG (Supplementary Fig. 17). TEM studies revealed that nanoribbons form 3D interconnected fibrous morphology (Fig. 3d). The AFM images of Zn-TPY-TTF CPG revealed the height of nanoribbon is ~7 nm and the diameter is in the range of 40–80 nm (Fig. 3f–h). The elemental mapping of the xerogel exhibited the uniform distribution of Zn$^{II}$ in a 3D network of the CPG (Supplementary Fig. 18). EDAX and elemental analyses also correlated the 2:1 ratio of Zn$^{II}$: TPY-TTF in the CPG (Supplementary Fig. 19). The high-resolution TEM analysis exhibited ordering in the nanoribbon, and lattice fringes were observed with a distance of 3.6 Å, which could be attributed to the intermolecular π–π stacking between the TTF---TPY units from the [Zn(TPY)$_2$]$^{2+}$ (Fig. 3e). Further, the PXRD pattern for Zn-TPY-TTF CPG xerogel showed a peak at $2\theta = 24.9°$ (3.6 Å), which was also observed in the gel state, justifying the presence of π–π stacking (Fig. 3k, Supplementary Fig. 20). In order to evaluate the nature of stacking, TD-DFT calculations were performed, which showed that the packing of TTF with [Zn(TPY)$_2$]$^{2+}$ on top of each other was stabilized with a distance of 3.56 Å (Fig. 3j, l) which is in good agreement with the experimental observations[50,54]. On the contrary, stacking through TTF---TTF units on the top of each other was optimized, which revealed the TTF---TTF distance >11 Å due to steric repulsion among the [Zn(TPY)$_2$]$^{2+}$ units attached to TTF core, and therefore, the possibility of TTF---TTF stacking was ruled out (Supplementary Fig. 22a). Further, the UV-vis absorption spectrum of Zn-TPY-TTF CPG in xerogel state was found to be similar to the TPY-TTF OG with an enhanced absorption in the visible region as shown in Fig. 3i. To analyse the reason behind enhanced absorption in the visible range, TD-DFT calculations were performed for the stacked model of Zn-TPY-TTF CPG (Fig. 4b). The result showed that the band

observed at 510 nm was similar to TPY-TTF OG. Whereas the experimental band at 565 nm is mainly attributed to the theoretical band at 553 nm (Fig. 4b, Supplementary Fig. 22b). Importantly, the transition at 553 nm was observed due to the significant contribution of intermolecular CT from TTF core to [Zn(TPY)$_2$]$^{2+}$ unit as shown in Supplementary Table 2[55]. The dominated intermolecular CT transition in Zn-TPY-TTF CPG as compared to TTF-TPY OG is mainly triggered by the planarization of terpyridine ligand, which occurred after the complexation with Zn$^{II}$ ion in CPG[56]. Additionally, Zn$^{II}$ complexation also increased the electron-accepting tendency of terpyridine ligand from the TTF moiety, which further facilitated the intermolecular CT transition[57]. The optical bandgap for Zn-TPY-TTF CPG was calculated to be 2.27 eV, which is closer to the bandgap of TPY-TTF OG (Supplementary Fig. 13). Next, as a controlled study, we found energies of the LUMO for TTF(PhCONH$_2$)$_4$ and [Zn(TPY)$_2$]$^{2+}$ are −1.91 and −2.17 eV, respectively, in the aqueous medium (Fig. 4c), which indicates that excited-state electron transfer is energetically favourable from TTF(PhCONH$_2$)$_4$ core to [Zn(TPY)$_2$]$^{2+}$ centre.

Mott-Schottky analysis was performed for the xerogel of both TPY-TTF OG and Zn-TPY-TTF CPG to evaluate experimental feasibility for water and CO$_2$ reduction (see SI for details). The M-S plots exhibited n-type nature with a positive slope for both TPY-TTF OG and Zn-TPY-TTF CPG (Fig. 5a). The flat band potentials ($V_{fb}$) were found to be −0.60 and −0.54 V versus RHE (at pH = 7) for TPY-TTF OG and Zn-TPY-TTF CPG, respectively. Based on the bandgaps obtained using UV-vis diffuse reflectance spectrometry (Supplementary Fig. 13), the electronic band structures versus RHE at pH 7 could be elucidated and are displayed in Fig. 5b[58]. Interestingly, the band alignments are shown in Fig. 5b illustrate that both TPY-TTF OG and Zn-TPY-TTF CPG possess suitable band edge positions to perform water and CO$_2$ reduction under visible-light irradiation.

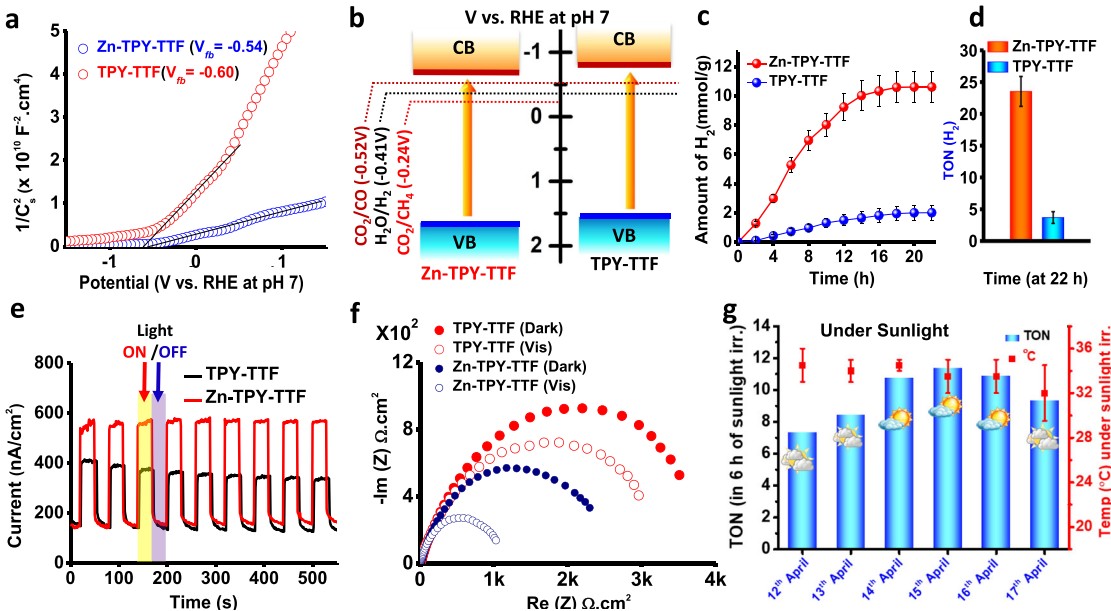

**Fig. 5 Electrochemical characterization and photocatalytic H₂ production performances for TPY-TTF OG and Zn-TPY-TTF CPG. a** Mott-Schottky plot w.r.t. RHE at pH = 7 (at 1000 Hz, −1.5 to +1.5 V). **b** Band alignment based on Mott-Schottky plot (V vs RHE at pH = 7; valence band (VB) and conduction band (CB)). **c** Amount of H₂ evolution under visible light and **d** corresponding TON (for H₂) at 22 h. (Error bars in panels **c** and **d** represents the standard deviations of three catalytic runs). **e** Photocurrent measurement in 0.5 M Na₂SO₄ at +0.8 V, pH ~7 upon visible-light irradiation in the time interval of 30 sec. **f** Nyquist plot under light and dark condition. **g** TON (for H₂) for Zn-TPY-TTF CPG under direct sunlight irradiation from 12th to 17th April 2019 (10:00 am to 4:00 pm) and the error bars represents the temperature range of photocatalytic set-up during catalysis under the sunlight irradiation. (Note: all visible-light experiments were performed under 300 W Xenon lamp using visible bandpass filter; 400–750 nm).

**H₂ production activity of OG and CPG under laboratory condition**. We have examined the potential of Zn-TPY-TTF CPG for photocatalytic H₂ production from water under visible-light (400–750 nm) irradiation using 300 W xenon lamp as the light source (Supplementary Figs. 23–25). The photocatalytic H₂ production activity of Zn-TPY-TTF CPG was screened with different sacrificial electron donors, and the best activity was found with triethylamine (TEA) as a sacrificial agent (Supplementary Fig. 26). Photocatalytic activity of Zn-TPY-TTF CPG was examined in both gel and xerogel state, and similar H₂ evolution was observed for both under similar conditions (Supplementary Fig. 27a). However, catalytic activities in different conditions were performed using the xerogel because of the ease of handling the catalyst in comparison to the gel state. After optimizing the catalyst loading (Supplementary Fig. 28), 1 mg of Zn-TPY-TTF CPG in xerogel state was dispersed in 38 ml of water for the photocatalytic H₂ production, and 2 ml triethylamine (TEA) was added into it, which acted as a sacrificial electron donor. The photocatalytic activity of Zn-TPY-TTF CPG was monitored by gas chromatography (GC) analysis and showed 10.60 mmol g⁻¹ of H₂ evolution in 20 h (activity = 530 μmol g⁻¹ h⁻¹) upon visible-light irradiation as shown in Fig. 5c. The amount of H₂ evolved was reached saturation in 20 h, and the turnover number (TON) was calculated to be 23.5, as shown in Fig. 5d. The activity is higher than many other transition metal-based photocatalysts (Supplementary Tables 6–8). The apparent quantum efficiency (AQE) for H₂ production using Zn-TPY-TTF CPG catalyst was calculated to be 0.76% at 550 ± 10 nm. Further, the absence of H₂ formation with Zn-TPY-TTF CPG under dark condition (absence of light) confirming light is an essential component for catalysis. Next, we have also performed the recyclability test by recollecting the catalyst followed by reusing the photocatalysis for four additional cycles of 6 h each time (Supplementary Fig. 27b). Interestingly, the amount of H₂ evolution was found to be similar in every cycle (>99%). We also performed ICP analysis for

as-synthesized Zn-TPY-TTF CPG sample as well as recovered sample after photocatalysis. In both cases, the amount of Zn content was calculated to be 5.9 (±0.2) wt%. Next, recycled catalyst (Zn-TPY-TTF) was collected at the end of the fourth cycle and analysed by FE-SEM and TEM studies and suggested no significant change in the structure and morphology after the catalytic reaction indicating high stability of the catalyst (Supplementary Fig. 27c, d).

Next, we have examined photocatalytic activity for the OG to compare the importance of morphology, i.e. the spatial arrangement of the chromophore and also the role of metal directed assembly in CPG. Experimental conditions employed for the TPY-TTF OG was similar to the CPG. Interestingly, the H₂ evolution by the TPY-TTF OG upon visible-light irradiation was increased with irradiation time and reached saturation in 22 h (Fig. 5c). The maximum H₂ evolution in 20 h was calculated to be 2 mmol g⁻¹ (activity = 100 μmol g⁻¹ h⁻¹), which is albeit lesser than the CPG but higher than most of the reported metal-free photocatalysts for H₂ evolution[59]. To understand the significant difference in photocatalytic activities between TPY-TTF OG and Zn-TPY-TTF CPG, photocurrent measurements were performed for both (Fig. 5e) in the presence and absence of light. The photocurrent for Zn-TPY-TTF CPG in the presence of light was found to be double as compared to TPY-TTF OG. This indicates the facile charge separation in Zn-TPY-TTF CPG under light irradiation; therefore, expected to show better photocatalytic activity as compared to TPY-TTF OG. This argument has been further validated by the EIS measurement, where the charge-transfer resistance for Zn-TPY-TTF CPG was observed to be significantly lesser as compared to TPY-TTF OG under both dark and light irradiated conditions (Fig. 5f). This can be attributed to the nanoribbon morphology of Zn-TPY-TTF CPG that provides a continuous charge-transfer pathway (via co-facial intermolecular charge delocalization) for the photogenerated electrons, which ultimately enhances the photocatalytic activity.

We have also performed the photocatalytic study with individual structural units of Zn-TPY-TTF CPG to evaluate the importance of coordination driven assembly of CPG in catalysis. Notably, individual components, such as TEA, $TTF(COOH)_4$, TPY-$NH_2$, and $[Zn(TPY-NH_2)_2]^{2+}$ as well as the physical mixture of $TTF(COOH)_4$ and $[Zn(TPY-NH_2)_2]^{2+}$ were found to be not efficient in catalysing $H_2$ evolution reaction (Supplementary Fig. 29). Next, the photocatalytic study has also been performed by making a blend of $Zn^{II}$ salt with TPY-TTF (in 2:1 ratio) (details are given in SI). This showed aggregated spherical morphology (sphere diameter $80 \pm 10$ nm) as confirmed by FE-SEM study (Supplementary Fig. 30), and the corresponding $H_2$ evolution within 6 h was found to be three times lesser as compared to the CPG (Supplementary Fig. 29). This signifies the impact of nano-structuring in photocatalytic performances. To validate the role of intermolecular CT interaction, we have also synthesized a coordination polymer of $Zn^{II}$ (Zn-CP) with $TTF(COOH)_4$ and characterized by FE-SEM, EDAX, elemental mapping, TGA, PXRD and UV-vis absorption study (Supplementary Figs. 31–34). Zn-CP showed micron-sized spherical particles with a diameter in the range of 2–3 μm. The Zn-CP showed 0.8 mmol $g^{-1}$ of $H_2$ production from water in 12 h which is eight times lesser in activity in comparison to Zn-TPY-TTF CPG photocatalyst under a similar condition (Supplementary Fig. 29). Overall, control experiments have unambiguously indicated that the coordination driven spatial arrangement of donor–acceptor chromophores and corresponding CT interaction have high significance in visible-light photocatalytic performances of CPG (Zn-TPY-TTF).

**Preparation and characterizations of Pt@Zn-TPY-TTF CPG.** Further, we envisioned that the entangled hierarchical fibrous structure of the coordination polymer gels could easily immobilize co-catalyst like Pt on the surface[60], which would facilitate the separation of photogenerated charge carriers by decreasing diffusion length and eventually enhance the photocatalytic activity[61]. Thus, we have successfully executed in situ generation and stabilization of platinum nanoparticles (Pt NPs) in the self-assembled interconnected network of Zn-TPY-TTF CPG (Pt@Zn-TPY-TTF) (Fig. 6a). High-resolution TEM and FE-SEM analysis have confirmed the stabilization of Pt NPs in self-assembled networks of Pt@Zn-TPY-TTF CPG within the size range of 2–3 nm (Fig. 6b, Supplementary Fig. 36a). Lattice fringes were observed for Pt NPs with the d-spacing value of 0.23 nm, indicating the presence of Pt (111) planes (Fig. 6c). Inductively coupled plasma-optical emission spectroscopy (ICP-OES) measurement indicated the presence of ~2.7 wt% Pt in Pt@Zn-TPY-TTF CPG, which is also supported by EDAX analysis (Supplementary Fig. 36b). The elemental mapping was ensured for the uniform distribution of Pt NPs in the gel matrix (Supplementary Fig. 37). Further, the Mott-Schottky analysis for Pt@Zn-TPY-TTF CPG also revealed that the conduction band edge occurs at $-0.51$ V vs RHE at pH 7, which is lesser compared to the Zn-TPY-TTF CPG ($-0.54$ V) catalyst (Supplementary Fig. 38a).

**$H_2$ production activity of Pt@Zn-TPY-TTF CPG.** Next, photocatalytic activity towards water reduction was examined for Pt@Zn-TPY-TTF CPG under a similar condition as employed for CPG as well as OG. Interestingly, Pt@Zn-TPY-TTF CPG has shown remarkably enhanced catalytic activity under visible-light irradiation, and the amount of $H_2$ was calculated to be 162.42 mmol $g^{-1}$ in only 11 h (average activity = 14727 μmol $g^{-1}$ $h^{-1}$), and the corresponding TON value was found to be 1176.9 (w.r.t. 2.7 wt% of Pt, Supplementary Table 3) as shown in Fig. 6d. We have also investigated

photocatalytic $H_2$ production activity by varying loading amount of Pt NPs, which displayed the highest catalytic activity in the presence of 2.7 wt% of Pt to the CPG (Supplementary Fig. 35). Drastically increased $H_2$ evolution after Pt NPs stabilization could be ascribed to the efficient charge separation in Pt@Zn-TPY-TTF CPG as Pt centre is a well-known electron acceptor that accumulates a pool of electrons and subsequently exhibits efficient water reduction. Here, the photoexcited electrons from Zn-TPY-TTF CPG using the harvested light energy transferred to Pt NPs surface to reduce water into $H_2$. Furthermore, the recyclability and reusability of Pt@Zn-TPY-TTF CPG towards photocatalytic hydrogen evolution were evaluated upto four cycles similar to the Zn-TPY-TTF CPG (Supplementary Fig. 39). Photocatalytic activity of the recycled catalyst was found to be retained >95% after 4th cycle. The formation of the Schottky junction in Pt@Zn-TPY-TTF CPG was helpful to separate the photogenerated electron-hole pairs. This argument was further validated by the EIS measurement, where the charge-transfer resistance for Pt@Zn-TPY-TTF CPG was found to be almost half as compared to Zn-TPY-TTF CPG under both dark and visible-light irradiation (Supplementary Fig. 38b). Furthermore, approximately four-folds higher photocurrent was observed for Pt@Zn-TPY-TTF CPG as compared to Zn-TPY-TTF CPG (Supplementary Fig. 38c), which corroborated the facile electron transfer from the Zn-TPY-TTF CPG to the Pt centre. Further, the photoluminescence (PL) spectra of Zn-TPY-TTF CPG showed weak emission with a maximum at 581 nm due to the intermolecular charge-transfer interaction ($\lambda_{ex} = 510$ nm). (Supplementary Fig. 40). PL spectra for Pt@Zn-TPY-TTF CPG upon excitation at 510 nm showed significantly quenched emission compared to Zn-TPY-TTF CPG. Therefore, to gain more insight into the advantages of Pt NPs in the increased charge separation, the time-resolved photoluminescence (TRPL) decay was studied on the Zn-TPY-TTF CPG and Pt@Zn-TPY-TTF CPG, as shown in Supplementary Fig. 41 (Supplementary Table 4). A higher average lifetime Zn-TPY-TTF CPG system (1.95 ns) compared to Pt@Zn-TPY-TTF CPG system (0.22 ns) obtained by the TRPL studies confirms that migration of photoexcited electrons is much faster in Pt@Zn-TPY-TTF compared to Zn-TPY-TTF CPG[62]. The decrease in the lifetime is attributed to the enhanced separation of photogenerated electrons in Pt@Zn-TPY-TTF CPG, which play decisive roles in enhancing the efficiency of photocatalytic processes. To understand the charge-separation dynamics, transient absorption (TA) experiments were performed with Zn-TPY-TTF CPG and Pt@Zn-TPY-TTF CPG. The TA spectra of Zn-TPY-TTF CPG (Supplementary Fig. 42a) showed a broad excited-state absorption (ESA) band covering 450–700 nm region at early time delay. Such a broad ESA band is due to TTF cation and TPY anion radical formed by the electron transfer process by the photoexcitation. TTF cation has an absorption band at ~440 nm with a long tail extending upto 700 nm[63], and Zn-TPY anion radical complex has a broad absorption band covering the range of 500–600 nm[64]. Thus, the appearance of a broad band at early time delay suggested that the electron transfer from TTF to Zn-TPY is very efficient in Zn-TPY-TTF CPG and takes place within the time resolution of the TA instrument (120 fs). The fitting of the decay trace at 570 nm due to TPY anion radical complex gives an average lifetime of 563 ps (Supplementary Fig. 42b). Pt@Zn-TPY-TTF CPG showed qualitatively similar TA spectra as of Zn-TPY-TTF CPG. However, the transient signal at 570 nm decays significantly faster in Pt@Zn-TPY-TTF CPG (average lifetime is 174 ps, see Supplementary Fig. 42c). Such faster decay kinetics in Pt@Zn-TPY-TTF CPG suggest electron transfer from reduced Zn-TPY to Pt NPs. Besides the TRPL and TA results,

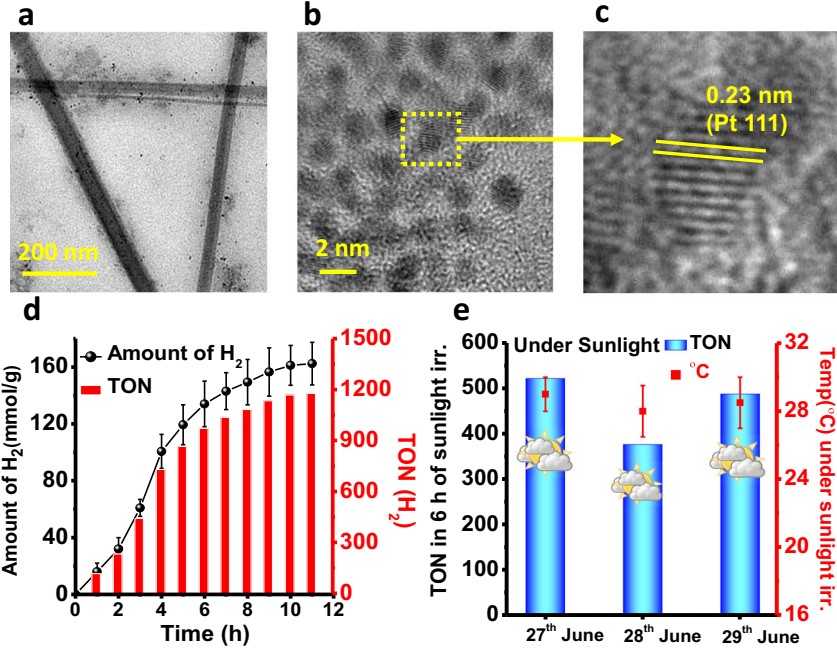

**Fig. 6 Characterizations and photocatalytic H₂ production activity of Pt@Zn-TPY-TTF CPG. a**, **b** HR-TEM images of Pt@Zn-TPY-TTF CPG. **c** Lattice fringes of the selected region. **d** Amount and TON (for H₂) under visible-light irradiation and error bars represents the standard deviations of three catalytic runs. **e** TON (for H₂) under direct sunlight irradiation from 27th June to 29th June 2019 (10:00 am to 4:00 pm) and the error bars represents the temperature range of photocatalytic set-up during catalysis under the sunlight irradiation.

DFT calculations also revealed that in Pt@Zn-TPY-TTF CPG, the electron transfer could take place from TTF to Pt NPs via $[Zn(TPY)_2]^{2+}$ unit. We have investigated the possible loading positions of Pt NPs in the CPG theoretically and the details have been given in SI (Supplementary Figs. 43–44). We were able to perform the stabilization energy calculations upto four atoms Pt cluster due to computational limitation (Supplementary Fig. 44) and detailed experimental study related to Pt NPs stabilization will be carried out in future works. Furthermore, AQE for the water reduction to H₂ was determined for the Pt@Zn-TPY-TTF CPG upon irradiating with monochromatic light of the wavelength of $400 \pm 10$ nm, $450 \pm 10$ nm, $500 \pm 10$ nm, $550 \pm 10$ nm, $600 \pm 10$ nm, $650 \pm 10$ nm, and $700 \pm 10$ nm (Supplementary Table 5, Supplementary Fig. 45). Notably, the highest AQE was obtained to be 14.47% at $550 \pm 10$ nm. These experiments suggested that the photocatalytic activity is mainly driven through intermolecular charge-transfer interaction. The H₂ evolution using Pt@Zn-TPY-TTF CPG was examined under both light and dark conditions (Supplementary Fig. 39c). No H₂ evolution was detected under dark condition, indicating the importance of light for water reduction. Next, photocatalysis was also performed for Pt@Zn-TPY-TTF and Zn-TPY-TTF CPG without any sacrificial donor (TEA). The aqueous dispersion of Zn-TPY-TTF CPG produced 0.92 mmol g⁻¹ of H₂ in 20 h, which is 11 times lesser than with TEA. Similarly, Pt@Zn-TPY-TTF CPG showed 30 times lesser activity without TEA (Supplementary Table 6).

**CO₂ reduction activity of OG and CPG under laboratory condition**. As mentioned above, the theoretical and experimental bandgap alignment of Zn-TPY-TTF CPG and TPY-TTF OG has the potential to reduce CO₂ as well. Therefore, we have performed visible-light-driven photocatalytic CO₂ reduction with the xerogel of Zn-TPY-TTF CPG and also compared it with TPY-TTF OG. TEA was used as a sacrificial electron donor for CO₂ reduction. First, screening of the solvent composition for CO₂

reduction has been performed (Supplementary Figs. 47–48, Supplementary Table 10), and the mixture of acetonitrile: water (3:1) have shown the best activity which could be attributed to high solubility of CO₂ in acetonitrile, whereas, water acted as a proton source[65]. Visible-light-driven CO₂ reduction by Zn-TPY-TTF CPG yielded 3.51 mmol g⁻¹ of CO in 8 h with >99% selectivity (activity = 438 µmol g⁻¹ h⁻¹) as shown in Fig. 7b. The AQE of CO₂ photoreduction for Zn-TPY-TTF CPG at $550 \pm 10$ nm was calculated to be 0.96%. Such a selective CO production is noteworthy and one of the best results among various reported hybrid photocatalysts systems (Supplementary Table 12). The TON for Zn-TPY-TTF CPG in 8 h was calculated to be 7.8 (Fig. 7b). Recyclability test for the Zn-TPY-TTF CPG photocatalyst was performed for four cycles, similarly as employed for the water reduction. The photocatalytic performances of Zn-TPY-TTF CPG were found to be almost unchanged (Supplementary Fig. 49). Further, the photocatalytic activity of Zn-TPY-TTF CPG was examined upon isotopic labelling with ¹³CO₂. This showed the formation of ¹³CO, which confirms that the produced CO was originated from CO₂ (Supplementary Figs. 50–51). Next, visible-light-driven photocatalytic CO₂ reduction has also been investigated for TPY-TTF OG under a similar condition as employed for Zn-TPY-TTF CPG. The TPY-TTF OG has displayed 1.12 mmol g⁻¹ CO formation in 11 h with >99% selectivity (activity = 140 µmol g⁻¹ h⁻¹), and the corresponding TON was estimated to be 2.1 as shown in Fig. 7a.

We have performed in situ Diffuse Reflectance Infrared Fourier Transform (DRIFT) spectroscopic study for Zn-TPY-TTF CPG to track the reaction intermediates formed in the course of CO₂ reduction to CO (Fig. 8a)[58]. Two peaks that appeared at 1514 and 1692 cm⁻¹ were assigned for the COOH* and COO⁻* intermediates, respectively, which are the signature intermediates formed during CO₂ reduction process (Fig. 8a)[66]. Peak observed at 1454 cm⁻¹ could be attributed to symmetric stretching of the HCO₃⁻* [67]. A noteworthy peak at 2074 cm⁻¹ was indicated for the formation of the CO*. Most importantly, the peak intensity of

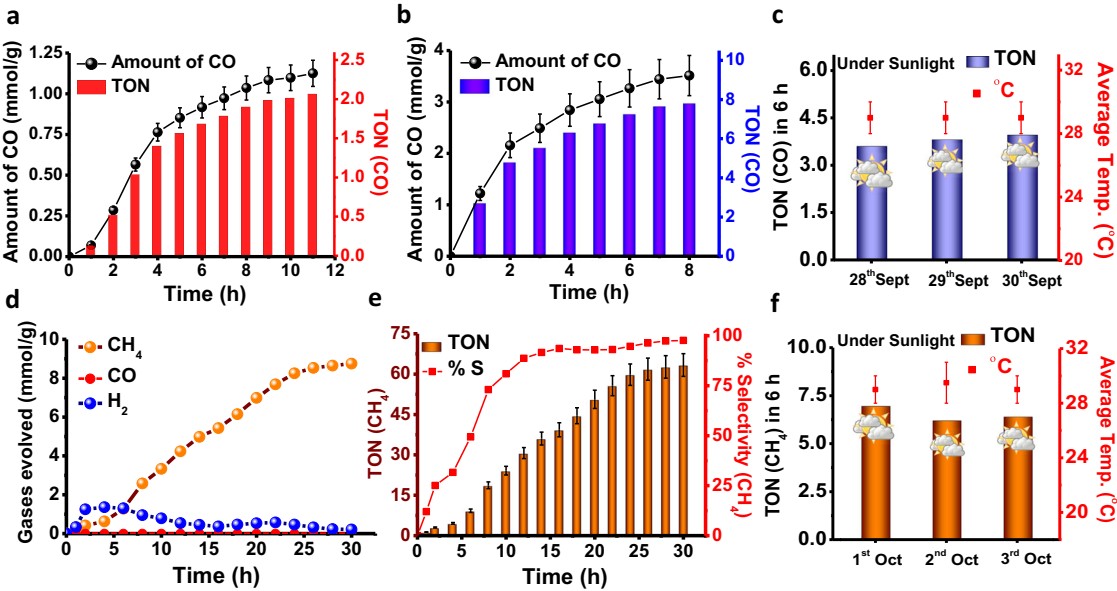

**Fig. 7 Photocatalytic activity towards CO₂ reduction. a** Amount and TON for CO formation in the presence of TPY-TTF OG under visible light. **b** Amount and TON for CO formation in the presence of Zn-TPY-TTF CPG under visible light. **c** TON for CO formation under sunlight from 28th to 30th Sept 2019 (10:00 am to 4:00 pm) in the presence of Zn-TPY-TTF CPG. **d** CO₂ reduction by Pt@Zn-TPY-TTF CPG (amount of CH₄ and H₂ formation with time) under visible light. **e** TON value and % selectivity for CH₄ formation using Pt@Zn-TPY-TTF CPG under visible light. **f** TON for CH₄ formation from CO₂ upon sunlight irradiation from 1st to 3rd Oct 2019 (10:00 am to 4:00 pm) in the presence of Pt@ Zn-TPY-TTF CPG. Error bars in panels **a**, **b** and **e** represents the standard deviations of three catalytic runs. Error bars in panels **c** and **f** represents the temperature range of photocatalytic set-up during catalysis under the sunlight irradiation.

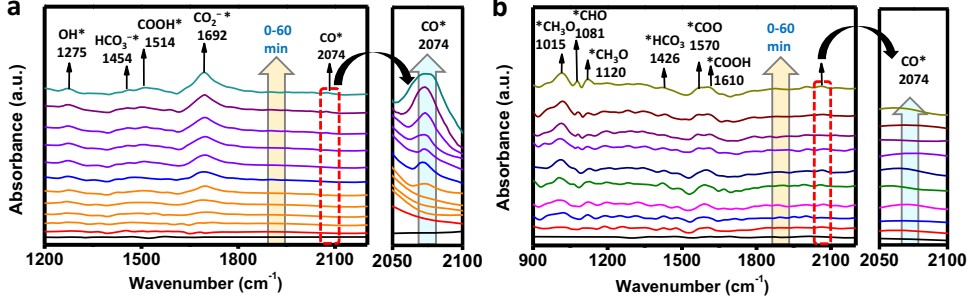

**Fig. 8 Real-time photocatalytic CO₂ reduction monitoring through in situ DRIFT study. a** For CO formation by Zn-TPY-TTF CPG. **b** For CH₄ formation by Pt@Zn-TPY-TTF CPG.

the CO* intermediate was substantially increased with reaction progress, suggesting CO* formation increases with time. Based on the experimental results and in situ DRIFT study, we have computed a plausible mechanism for CO₂ reduction, which is also in good agreement with the earlier report[68] (Supplementary Figs. 65–66). The photocatalytic cycle was initiated by the light absorption due to CT interaction in Zn-TPY-TTF CPG, and electron transfer took place from the excited state of TTF* to [Zn(TPY)₂]²⁺ units followed by reductive quenching of TTF⁺ by TEA. As a result, the [Zn(TPY)₂]²⁺ converted to the radical cation species [Zn(TPY⋅⁻)(TPY)]⁺. Notably, the electron spin density distribution plot revealed that the electron was localized at the terpyridine unit of the catalyst that resulted in [Zn(TPY⋅⁻)(TPY)]⁺ species with the stabilization energy of 0.52 eV. (Supplementary Fig. 65d and Supplementary Table 35). Next, as the reaction progress, the acetonitrile (CH₃CN), solvent molecule replaced one ligating site of terpyridine to afford [Zn(TPY⋅⁻)(η²-TPY)(CH₃CN)]⁺ which is slightly uphill (ΔG = +0.77 eV) with a low activation barrier of 0.81 eV. This [Zn(TPY⋅⁻)(η²-TPY)(CH₃CN)]⁺ intermediate subsequently binds with CO₂ molecule

with a distance of 2.41 Å to Zn^II ion and yielded the [Zn(TPY) (η²-TPY)(COO⁻)]⁺ species (ΔG = +0.46 eV). The formulation of the intermediate as [Zn(TPY)(η²-TPY)(COO⁻)]⁺ rather than [Zn(TPY⋅⁻)(η²-TPY)(COO)]⁺ is supported by the spin density distribution plot (Supplementary Table 38). Notably, the CO₂ molecule acted as a monodentate ligand, and the ∠O–C–O angle was found to be 139.36°, keeping in mind that the free CO₂ has linear geometry (Supplementary Fig. 65f and Supplementary Table 38). Here, it is worth mentioning that the one-electron charging over the [Zn(TPY)₂]²⁺ complex favours the binding of CO₂ molecule by increasing electron density around the metal centre, which is a prerequisite for the nucleophilic attack to the CO₂. However, the complex [Zn(TPY)(η²-TPY)(COO⁻)]⁺ was further reduced to form the singlet species [Zn(TPY⋅⁻)(η²-TPY) (COO⁻)] (ΔG = −0.79 eV) (Supplementary Fig. 65g and Supplementary Table 39). Next, the carboxylate centre of the singlet complex gets readily protonated and results in the formation of [Zn(TPY⋅⁻)(η²-TPY)(COOH)]⁺ complex (ΔG = −2.88 eV) (Supplementary Fig. 65h and Supplementary Table 40). In the next step, subsequent protonation and water elimination

from $[Zn(TPY^{\bullet-})(\eta^2\text{-}TPY)(COOH)]^+$ intermediate lead to the formation of $[Zn(TPY)(\eta^2\text{-}TPY)(CO)]^{2+}$ which is found to be a highly downhill process ($\Delta G = -2.55$ eV). (Supplementary Fig. 65i and Supplementary Table 41). In this complex, the CO molecule is loosely attached to the $Zn^{II}$ centre at a distance of 2.66 Å. As a result, the CO molecule can be easily released from the metal centre. Thus, the intermediate $[Zn(TPY)(\eta^2\text{-}TPY)(CO)]^{2+}$ gets readily reduced, leading to the subsequent removal of CO ($\Delta G = -1.51$ eV) which regenerates the active species $[Zn(TPY^{\bullet-})(TPY)]^+$ and re-enters into the catalytic cycle. Specifically, in this photocatalytic $CO_2$ reduction mechanism, the Zn-terpyridine complex involves retention of the bis-terpyridine ligation to gain the original coordination environment similar to as-synthesized Zn-TPY-TTF CPG.

**$CO_2$ reduction activity of Pt@Zn-TPY-TTF CPG**. Next, the photocatalytic activity of Pt@Zn-TPY-TTF CPG towards $CO_2$ reduction has also been examined under a similar experimental condition as applied for CPG and OG. Pt@Zn-TPY-TTF CPG showed excellent $CO_2$ reduction activity, and more interestingly, $CH_4$ was produced rather than CO as obtained for Zn-TPY-TTF CPG (Fig. 7d). It has already been reported in the literature that the presence of Pt NPs on the surface of semiconductor plays a key role in the formation of $CH_4$[60]. The formation of $CH_4$ has reached saturation in 30 h, and the corresponding yield was calculated to be 8.74 mmol $g^{-1}$ (activity = 292 µmol $g^{-1}$ $h^{-1}$) (Fig. 7d). During $CO_2$ reduction, a small amount of $H_2$ evolution was also observed ($\sim 0.20$ mmol $g^{-1}$ in 30 h), which was more in the initial hours but significantly decreased as the reaction progressed with time. Thus, the selectivity of $CO_2$ reduction to $CH_4$ formation in 30 h was noted to be more than 97% (Fig. 7e). The TON for $CH_4$ was calculated to be 63.4 (w.r.t. Pt loading on Zn-TPY-TTF CPG; Fig. 7e and Supplementary Table 3) in 30 h, which is better than many of the earlier reports (Supplementary Table 12). The recyclability of the catalyst, Pt@Zn-TPY-TTF CPG was examined for 6 h upto four cycles, and the amount of $CH_4$ formation in the fourth cycle was found to be >95%, indicating high stability of the catalyst (Supplementary Fig. 56). The ICP-OES analysis of Pt@Zn-TPY-TTF CPG was also performed after the fourth catalytic cycle, which showed 2.52 wt% of Pt present in the recovered catalyst. The partial decrease in catalytic activity (<5%) could be attributed to the minimal loss of Pt NPs during the recycling process. The AQE of the Pt@Zn-TPY-TTF CPG towards $CO_2$ reduction was calculated at different wavelengths using monochromatic lights (Supplementary Fig. 54 and Supplementary Table 9). The highest AQE for the $CH_4$ formation was obtained as 0.81% at 550 ± 10 nm, which further confirms that the photocatalytic activity of Pt@Zn-TPY-TTF is attributed to the intermolecular charge-transfer interactions. The $CO_2$ reduction was also performed under both light and dark conditions using Pt@Zn-TPY-TTF (Supplementary Fig. 55). The $CH_4$ formation was not increased under dark condition, justifying the importance of light in photocatalysis. Further, photocatalysis was performed with labelled $^{13}CO_2$ (isotopic labelling) using Pt@Zn-TPY-TTF (Supplementary Figs. 52–53). This showed the formation of labelled $^{13}CH_4$ and confirming that $CO_2$ is the actual source for the $CH_4$ formation. Next, we have performed in situ DRIFT experiment for Pt@Zn-TPY-TTF in order to monitor the real-time progress of the $CO_2$ reduction reaction (Fig. 8b). IR-stretching peaks observed at 1610, 1570, and 1426 $cm^{-1}$ could be attributed to the intermediates $COOH^*$, $COO^*$, and $HCO_3^*$, respectively[58]. Weak intensity peak for the $CO^*$ at 2062 $cm^{-1}$ illustrated that the $CO^*$ could be readily converted to other multi-electron reduction intermediates. Moreover, the characteristic intermediates for $CH_4$ formation were observed at 1081 $cm^{-1}$

($CHO^*$), 1015 and 1120 $cm^{-1}$ ($CH_3O^*$)[58]. Based on the DRIFT study, the plausible mechanism for the $CH_4$ formation was elucidated with the help of free energy calculation of different intermediates using DFT, which suggested that the Pt NPs is likely to be the catalytic centre during the $CO_2$ reduction (Supplementary Figs. 67–70).

**Sunlight-driven $H_2$ production**. The above discussions have clearly shown that visible-light photocatalytic activity and stability of both Zn-TPY-TTF CPG and Pt@Zn-TPY-TTF CPG in the xerogel state is indeed impressive. Notably, the amount of $H_2$ evolution using photocatalyst Zn-TPY-TTF CPG is higher than previously reported non-precious metal-based photocatalyst materials (Supplementary Table 8). Therefore, our next goal was to examine the potential of Zn-TPY-TTF towards $H_2$ evolution upon direct sunlight irradiation at ambient condition (Supplementary Fig. 24). The experiment was performed with Zn-TPY-TTF CPG under sunlight from 10:00 am to 4:00 pm for one week from 12th to 17th April 2019 on the rooftop of our institute. The weather condition corresponding to the above-mentioned period can easily be found out on the web (www.timeanddate.com). Interestingly, maximum $H_2$ evolution of 5.14 mmol $g^{-1}$ in 6 h (activity = 857 µmol $g^{-1}$ $h^{-1}$) was observed on 15th April, which is comparable with the amount of $H_2$ obtained under laboratory conditions (Xe-lamp irradiation). The TON for $H_2$ evolution was calculated for the above-mentioned period (Fig. 5g). The highest TON value of 11.9 was obtained on 15th April 2019. Whereas the lowest TON was calculated to be 7.2 on 12th April 2019 because of partially cloudy weather (Supplementary Table 7). Next, similar to Zn-TPY-TTF CPG, we have also examined sunlight-driven photocatalytic $H_2$ evolution for Pt@Zn-TPY-TTF CPG in xerogel state (Fig. 6e). The experimental condition for Pt@Zn-TPY-TTF CPG was similar to the Zn-TPY-TTF CPG. Nevertheless, experimental timing was different for the Pt@Zn-TPY-TTF CPG. The experiment with Pt@Zn-TPY-TTF CPG was performed from 27th to 29th June 2019. The highest $H_2$ evolution was calculated to be 72 mmol $g^{-1}$ in 6 h (activity = 12,000 µmol $g^{-1}$ $h^{-1}$) on 27th June 2019, and the corresponding TON was calculated to be 521.8 (w.r.t. Pt loading), which is indeed noteworthy (Fig. 6e; Supplementary Table 3). Whereas the lowest TON value for $H_2$ evolution was found to be 376.48 (w.r.t. Pt loading) on 28th June due to partially cloudy weather.

**Sunlight-driven $CO_2$ reduction**. Interestingly, the potential of Zn-TPY-TTF CPG for $CO_2$ reduction has also been examined under direct sunlight between 10:00 am to 4:00 pm for three days from 28th to 30th September 2019. The highest CO formation of 1.79 mmol $g^{-1}$ was observed in 6 h (activity = 298 µmol $g^{-1}$ $h^{-1}$) on 30th September 2019, and the corresponding TON was calculated to be 3.9 (Fig. 7c). Next, sunlight-driven $CO_2$ reduction has been performed with Pt@Zn-TPY-TTF CPG for three days from 1st to 3rd October 2019 (Fig. 7f). Similar to the laboratory conditions, Pt@Zn-TPY-TTF CPG upon sunlight irradiation has displayed $CH_4$ formation (Supplementary Table 11). The highest $CH_4$ formation of 0.96 mmol $g^{-1}$ in 6 h (activity = 160 µmol $g^{-1}$ $h^{-1}$) was observed on 1st October 2019, and the corresponding TON value was calculated to be 6.9 (w.r.t. Pt loading; Supplementary Table 3). Whereas the lowest $CH_4$ evolution with the TON value of 6.24 in 6 h (w.r.t. Pt loading) took place on 2nd October 2019. The CO or $CH_4$ formation under direct sunlight irradiation is albeit lower than the laboratory condition, but the obtained amount under ambient condition is quite exciting and promising because of practical application. Furthermore, after performing sunlight-driven photocatalysis with Zn-TPY-TTF CPG and Pt@Zn-TPY-TTF CPG catalysts were recovered

through centrifugation and washed with fresh water 3–4 times. The FE-SEM and TEM analysis, PXRD, FT-IR and UV-vis absorption experiments were performed for the recovered catalysts and found to be similar to the as-synthesized material, indicating that the structural integrity of the material remained intact after photocatalysis (Supplementary Figs. 57–64). Whereas the EDAX analysis ensured the presence of all the elements in similar quantity as obtained for as-synthesized Zn-TPY-TTF and Pt@Zn-TPY-TTF CPG (Supplementary Figs. 57 and 61). This unambiguously demonstrated the high stability of these catalysts during sunlight irradiation.

## Discussion

In a nutshell, we have successfully demonstrated a TTF-based soft processable metal-organic hybrid gel as a visible-light photocatalyst for $H_2$ evolution and $CO_2$ reduction to carbonaceous fuel such as CO/CH$_4$. Here, charge-transfer-driven photocatalysis based on coordination polymer gel has been exploited, where earth-abundant metal ions play a crucial role in the spatial organization of donor–acceptor π-chromophores to drive the catalytic activity. Further, we have shown the catalytic activity of the CPG after decorating Pt NPs as co-catalyst. It enhances the rate of $H_2$ production in 20 folds and dramatically changes the $CO_2$ reduction product from CO to CH$_4$. We have also demonstrated efficient catalytic activity of the CPG and Pt decorated CPG under sunlight with high selectivity. The real-time reaction progress of $CO_2$ reduction was monitored by DRIFT studies, and based on that, a plausible mechanism of $CO_2$ reduction was elucidated for CPG. The easy processability and structural tunability of LMWG offers a lot of room for designing efficient photocatalyst materials for practical application. Our work will pave the way toward designing coordination driven hybrid soft processable photocatalyst systems for solar energy driven fuel production.

## Methods

**Reagent**. Tetrathiafulvalene (TTF), 1,3-diaminopropane, 4′-chloro-2,2′:6′,2″-terpyridine, Zinc nitrate (Zn(NO$_3$)$_2$.6H$_2$O), Ethyl-4-bromobenzoate (Br-C$_6$H$_4$COOEt), Cesium carbonate (Cs$_2$CO$_3$), Palladium acetate, Tri-tertbutyl-phosphonium tetrafluoroborate (PtBu$_3$.HBF$_4$), Thionyl chloride (SOCl$_2$), Chloroplatinic acid (H$_2$PtCl$_6$.6H$_2$O) were purchased from Sigma–Aldrich chemical Co. Ltd. Spectroscopic grade solvents were used for all spectroscopic studies without further purification.

**Synthesis of TPY-TTF LMWG**. TTF(COOH)$_4$ (634 mg, 1.65 mmol) was dissolved in 50 ml of dry tetrahydrofuran (THF), and thionyl chloride (SOCl$_2$) (2.4 ml, 33 mmol) was added into it under inert condition. The reaction mixture was refluxed for 2 h at 65 °C. Then the reaction mixture was distilled at 120 °C to remove excess SOCl$_2$ and yielded a solid precipitate of acid chloride. Next, the solid precipitate was dissolved in 40 ml of dry THF. Now the solution of TPY-NH$_2$ (2.21 g, 7.26 mmol) in 10 ml of dry THF along with triethylamine (1.25 ml, 9 mmol) was added to the solution of acid chloride dropwise at 0 °C. The reaction was stirred at 0 °C for 12 h. The solid precipitate was formed, which was filtered and washed with chloroform and acetone to remove unreacted TPY-NH$_2$. The yield of isolated dark red solid precipitate (TPY-TTF LMWG) was found to be 28%. $^1$H-NMR (600 MHz, DMSO -$d_6$): $\delta$ = 8.80 (d, 8H), 8.71 (broad, 4H), 8.10 (m, 8H), 8.005 (d, 8H), 7.83 (s, 8H), 7.58 (m, 8H), 7.48 (d, 8 H), 7.11 (m, 8H), 4.42 (m, 4H), 3.52 (m, 8 H), 3.44 (m, 8H), 1.93 (m, 8H). $^{13}$C NMR {$^1$H} (150 MHz, DMSO -$d_6$): $\delta$ = 166.91, 156.25, 155.75, 155.32, 149.27, 137.42, 135.72, 131.72, 130.28, 129.52, 129.44, 127.29, 124.26, 120.98, 108.05, 37.22, 36.67, 26.82. Selected FT-IR (KBr, cm$^{-1}$): 3376 (b), 3232 (w), 3036 (s), 2930 (w), 1635 (s), 1573 (s), 1475, 1362, 1301 (w), 1102 (w), 990 (w), 849(w), 786 (s), 611(b). MALDI-TOF (m/z): [M]$^+$ Calcd. for C$_{106}$H$_{88}$N$_{20}$O$_4$S$_4$, 1834.22; found [M + H]$^+$, 1835.06; analysis (Calcd., found for C$_{106}$H$_{88}$N$_{20}$O$_4$S$_4$): C (69.81, 69.41), H (4.82, 4.84), N (15.25, 15.27), S (6.78, 6.99).

**Preparation of TPY-TTF OG**. TPY-TTF (10 mg) was dissolved in a mixture of MeOH, DCM and H$_2$O (2:1:1 ratio) (300 µl) and sonicated for 2–3 min and then heated at 60 °C to get a homogenous viscous solution. The mixture was cooled to room temperature and kept for 2 h, which has resulted in an opaque organogel (TPY-TTF OG). The formation of the gel was confirmed by rheology measurements. Further, xerogel of TPY-TTF OG was synthesized by heating the gel at

80 °C under vacuum for 8 h. Selected FT-IR (KBr, cm$^{-1}$) for xerogel state of TPY-TTF OG: 3392 (b), 3051 (s), 2945 (w), 1649 (s), 1581 (s), 1460 (s), 1362 (w), 1294–1096 (w), 976 (w), 839 (s), 786 (s), 619 (b).

**Preparation of Zn-TPY-TTF CPG**. We have used Zn(NO$_3$)$_2$.6H$_2$O salt to synthesize Zn-TPY-TTF CPG. TPY-TTF (10 mg, 5 µmol) was taken in the 300 µl solvent mixture of MeOH, DCM and H$_2$O (2:1:1 ratio), and 10 µmol of Zn$^{II}$ was added into it at 60 °C. The reaction mixture was heated for a few minutes to get a viscous solution and kept for 4 h at room temperature, which transformed into an opaque gel. The formation of Zn-TPY-TTF gel was confirmed by rheology test. Further, xerogel of Zn-TPY-TTF CPG was synthesized by heating the gel at 80 °C under vacuum for 8 h. Selected FT-IR for Zn-TPY-TTF CPG (KBr, cm$^{-1}$): 3414 (b), 2922 (s), 2672 (s), 2490 (s), 1619 (s), 1468 (s), 1377 (w), 1233–1021 (w), 791 (s), 629 (b), 536 (b). Analysis (Calcd., found for C$_{106}$H$_{88}$N$_{24}$O$_{16}$S$_4$Zn$_2$): C (57.48, 57.34), H (4.10, 3.96), N (15.18, 15.12), S (5.79, 5.75).

**Preparation of Pt@Zn-TPY-TTF CPG**. For in situ platinum nanoparticles (Pt NPs) stabilization, 1 mg of H$_2$PtCl$_6$·6H$_2$O was dispersed in 40 ml of water containing 10 mg of Zn-TPY-TTF CPG. After continuous stirring for 1 h in a closed system, the well-dispersed solution was irradiated using 300 W xenon lamp (Newport) with a 6.0 cm long IR water filter for 2 h. Finally, the Pt nanoparticle stabilized CPG sample (Pt@Zn-TPY-TTF) was thoroughly washed with deionized (DI) water and dried under vacuum at 80 °C for 12 h. More detailed characterization of Pt@Zn-TPY-TTF CPG is provided in Supplementary Information.

## Characterization

*General*. $^1$H-NMR spectra were recorded on a Bruker AVANCE-400 NMR spectrometer (at 400 MHz) and JEOL-ECZR NMR spectrometer (at 600 MHz) with chemical shifts recorded as ppm, and all spectra were calibrated against TMS. $^{13}$C-spectrum was recorded at 150 MHz frequency using a Varian Inova 600 MHz NMR spectrometer. UV-Vis spectra were recorded in a Perkin-Elmer Lambda 900 spectrometer. For the UV-vis absorption studies, TPY-TTF OG and Zn-TPY-TTF CPG and Pt@Zn-TPY-TTF CPG were coated on a quartz plate as a thin film. Time-resolved photoluminescence (TRPL) and photoluminescence (PL) studies were performed on an Edinburgh instrument (FLS 1000). Fourier transform infrared spectra (FT-IR) were recorded by making KBr pellets using Bruker IFS 66 v/S Spectrophotometer in the region 4000–400 cm$^{-1}$. Thermal stability of the materials was studied using Mettler Toledo TGA 850 instrument in the temperature range of 30–800 °C with the heating rate of 5 °C/min in N$_2$ atmosphere. Powder X-ray diffraction (PXRD) patterns were measured by a Bruker D8 Discover instrument using Cu Kα radiation. Atomic force microscopy (AFM) measurements were carried out with a Nasoscope model Multimode 8 Scanning Probe Microscope to analyze the morphologies of the sample surface. For this analysis, samples were dispersed in ethanol and then coated on Si wafer by a drop-casting method. The Field Emission Scanning Electron Microscopic (FE-SEM) images, elemental mapping, and Energy-dispersive X-ray spectroscopy (EDAX) analysis were recorded on a Nova Nanosem 600 FEI instrument. The xerogels were dispersed in ethanol and then drop-casted onto a small piece of silicon wafer followed by gold (Au) sputtering for FE-SEM measurements. Transmission Electron Microscopy (TEM) studies were done on JEOL JEM -3010 with an accelerating voltage of 300 kV. For this analysis, the xerogels were dispersed in ethanol and drop cast on a carbon copper grid. Elemental analyses were carried out using a Thermo Scientific Flash 2000 CHN analyzer. MALDI was performed on a Bruker daltonics Autoflex Speed MALDI-TOF System (GT0263G201) spectrometer. High-resolution mass spectrometry was carried out using Agilent Technologies 6538 UHD Accurate-Mass Q-TOFLC/MS. Metal contents in the CPGs were estimated by Inductively coupled plasma-optical emission spectrometry (ICP-OES) on Perkin-Elmer Optima 7000dv ICP-OES. For the determination of Zn and Pt, CPG samples were digested with HNO$_3$ and HCl and analysed by ICP-OES.

*Rheology*. The rheological study was done in Anton Paar Rheometer MCR 302. Rheological measurements were operated in a 25 mm cone-and-plate configuration with a 0.5° cone angle. The rheology experiment was performed using the amplitude sweep method over strain % at a constant frequency (ω = 1 Hz). For each rheology measurement, the gel was prepared in 10 ml glass vial. Next, ~20 mg of gel sample was loaded onto the rheometer plate with the help of a spatula in a single shot to avoid any damage to the loaded sample. Further, data accuracy was ensured by repeating these experiments a minimum of three times.

*Critical point drying*. Tousimis Autosamdri@931 was used for critical-point drying (CPD) of the gel samples. After gel preparation, solvents present in CPG, were exchanged with ethanol using a gradient of ethanol/water mixtures (40–100 %). Next, the ethanol exchanged gel samples were then transferred to a stainless-steel cage with wire mesh followed by critically point dried with supercritical CO$_2$.

*Transient absorption*. Femtosecond transient absorption (TA) experiments were performed using an amplified Ti:sapphire laser system (800 nm, 60 fs, 1 kHz, 3 mJ) from Amplitude Technologies, France, and a pump-probe set-up (Excipro) from CDP Corporation, Russia. A fraction of the fundamental laser beam was used to

generate 400 nm pump pulse in a 0.2 mm thick BBO crystal. Another small fraction of 800 nm pulse was used to generate broad continuum pulse (350–750 nm) in a rotating calcium fluoride window. The time delay between the pump and probe pulses was maintained by using an optical delay stage in the probe path. To minimize the noise in the transient signal, a part of the probe beam, known as reference pulse, was passed through the unexcited region of the sample and detected simultaneously with the probe pulse using a dual diode array-based spectrometer. The probe pulse was focussed in the sample and spatially overlapped with the focussed pump pulse. To obtained the pump induced changes in the absorbance of the sample ($\Delta A$), the alternate pump pulses were blocked with the help of a synchronized chopper. Each transient spectra were collected after averaging for 2 s at each delay time. Samples were dispersed in methanol and taken in a rotating cell to avoid their photodecomposition. The instrument response function of the TA set-up was 120 fs.

*Mott-Schottky.* The energy band structure of TPY-TTF OG and Zn-TPY-TTF CPG and Pt@Zn-TPY-TTF CPG was depicted by the Mott-Schottky (MS) analysis (at 1000 Hz, from −2.0 to +2.0 V) using ITO as a working electrode (WE) in $N_2$-purged aqueous solution of 0.5 M $Na_2SO_4$ at pH = 7, Pt as a counter electrode (CE) and Ag/AgCl as a reference electrode (RE). An electrochemical ink was prepared by making a dispersion of a mixture of catalyst (2.0 mg) in the solvent mixture of isopropanol (500 μl), water (500 μl), and Nafion (14 μl). Upon sonication for 20 min, a well-dispersed ink (3.5 μl) was drop cast over the ITO electrode and allowed to dry for 3 h under ambient condition.

*Photocurrent.* The similar set-up was used for photocurrent measurements as employed for Mott-Schottky analysis. Here, the photocurrent study was performed for TPY-TTF OG, Zn-TPY-TTF CPG, and Pt@ Zn-TPY-TTF CPG upon consecutive light "ON-OFF" cycles for 30 s over 10 cycles.

*Electrochemical impedance spectroscopy (EIS).* This experiment was performed in a three-electrode cell configuration with a glassy carbon electrode as the WE, platinum as a CE, and Ag/AgCl as a RE. 0.5 M $Na_2SO_4$ was used as an electrolyte at pH = 7. An electrochemical ink was prepared by making a dispersion of a mixture of catalyst (2.0 mg) in the solvent mixture of isopropanol (500 μl) and water (500 μl). Upon sonication for 30 min, a well-dispersed ink (3.5 μl) was drop cast over the GC electrode and allowed to dry for 3 h under ambient condition. EIS was recorded at −1.2 $V_{RHE}$ applied bias from 0.1 Hz to 100 kHz (under the dark condition and visible-light irradiation).

## Photocatalytic experiments

*Experimental set-up for photocatalytic water reduction.* Photocatalytic $H_2$ evolution experiments were carried out in an 80 mL self-designed borosilicate glass cell containing a magnetic stir bar sealed with a small septum (picture of photocatalytic cell is given in Supplementary Fig. 23). For photocatalytic experiment, 1 mg catalyst was dispersed in 38 mL water containing 2 ml of triethylamine (TEA) as a sacrificial agent. The suspension was ultrasonicated to make a homogeneous dispersion. The reaction mixture was then purged with $N_2$ for 30 min to remove any traces of dissolved $H_2$ gas, which was ensured by GC-analysis before performing the photocatalysis. The reaction mixture was irradiated with a 300 W Xenon lamp (Newport) fitted with a 12 cm path length of water filter for removal of IR radiation. A visible bandpass filter (400–750 nm) was used to block the UV light. The Headspace gases were sampled using Hamilton air-tight syringes by injecting 250 μL into the gas chromatograph (Agilent CN15343150). Gas Chromatography referencing was done against a standard ($H_2/N_2$) gas mixture with a known concentration of hydrogen for the calibration curve, where $N_2$ was used as a carrier gas, and a thermal conductivity detector (TCD) was used for $H_2$ detection (Supplementary Fig. 25). Notably, no hydrogen evolution was observed for a mixture of water/5 Vol % TEA under visible-light irradiation in the absence of a photocatalyst.

*Experimental set-up for the photocatalytic $CO_2$ reduction.* The photocatalytic $CO_2$ reduction reaction was carried out in a similar reaction vessel as discussed above for the water reduction. Notably, a mixture of acetonitrile ($CH_3CN$) and $H_2O$ in 3:1 ratio was used as a solvent for the $CO_2$ reduction. In short, 38 ml solvent mixture ($CH_3CN$: $H_2O$ in 3:1), 2 ml of TEA as a sacrificial electron donor, and 1 mg of the catalyst were taken in a reaction flask and dispersed uniformly through sonication. The reaction vessel was sealed with a septum and then purged with $CO_2$ of 99.9% purity for ~30 min in order to make $CO_2$ saturated atmosphere. The reaction mixture was irradiated with visible light as employed for water reduction. During the $CO_2$ reduction reaction, the gas in the headspace of the reaction vessel was analyzed qualitatively and quantitatively by GCMS-QP2020. During light exposure, the evolved gases in the headspace of the reaction vessel were collected by a hamilton syringe and injected in GC-MS at every 1 h time interval until product production ceased. The mass detector was used to analyze the mass of evolved products such as CO, $CH_4$, $CH_3OH$, HCOOH, and $CO_3^{2-}$. The $H_2$, CO, and $CH_4$ were detected by RT®-Msieve 5 A column (45 meters, 0.32 mmID, 30 mdf). To detect the HCOOH, Stabilwax®-DA (30 meters, 0.18 mmID, 0.18 μmdf) column was used, and for Methanol, SH®-Rxi-5Sil MS (30 meters, 0.25 mmID, 0.32 μmdf)

column was used in GCMS. The calibration was done by a standard gas mixture of $H_2$, CO, and $CH_4$ of different concentrations in ppm-level (Supplementary Fig. 46). Importantly, the GCMS has a detection limit of 1.0 ppm for $H_2$, CO, and $CH_4$. After the photocatalysis, the reaction mixture was filtered to remove the residual solid, and the solution was further analyzed to determine the amount of HCOOH/MeOH. All described data points are the average of at least 3 experiments. For isotopic labelling experiment, one litre $^{13}CO_2$ gas cylinder was purchased from Sigma–Aldrich (details: 99.0% ATOM % $^{13}C$, <3 Atom % $^{18}O$; M.W. 45.00 g/mol). We have purged the $^{13}CO_2$ for 10 min in a controlled manner to the photocatalytic reaction mixture of Zn-TPY-TTF CPG as well as Pt@Zn-TPY-TTF CPG.

*In situ diffuse reflectance infrared Fourier transform (DRIFT) spectroscopy measurements.* The in situ DRIFT measurements were carried out by FT-IR spectrophotometer (BRUKER, Pat. US, 034, 944) within a photoreactor. The 6 mg of catalyst was evenly spread over a glass disc of 1 cm diameter and placed inside the photoreactor for monitoring the reaction progress of photocatalytic $CO_2$ reduction. Next, the air was removed using vacuum inside the cell, and then 99.99% $CO_2$ gas along with water vapour was passed for 15 min inside the photoreactor. At last, the visible light was irradiated on catalyst by 150 W white LED light (>400 nm). In situ FT-IR signal was collected through MCT detector at regular time interval.

*Lifetime calculations.* Lifetime data for Zn-TPY-TTF CPG and Pt@Zn-TPY-TTF CPG were collected upon exciting at 510 nm. The average lifetime is calculated using following formula.

$$\tau_{avg} \text{ (ns)} = \frac{\Sigma A_i \tau_i^2}{\Sigma A_i \tau_i} \tag{1}$$

Where, $\tau_{avg}$ = average lifetime in nano-seconds, $\Sigma A_i$ = sum of percentage of all the components that exist in the excited state, $\Sigma \tau_i$ = sum of excited-state lifetime of all the component. The details are provided in Supplementary Table 4.

*Turnover Number.* The turnover number (TON) is calculated by using the below formula 2.

$$\text{TON} = \text{Amount of product evolved } (\mu mol)/\text{amount of catalyst } (\mu mol) \tag{2}$$

For hydrogen production. 1.0 mg of TPY-TTF OG is equivalent to 0.545 μmol and the amount of $H_2$ evolved was found to be 2 μmol from 1 mg of TPY-TTF in 20 h. Therefore, TON was calculated for TPY-TTF OG in 20 h is 3.6. The unit of Zn-TPY-TTF CPG was calculated based on a binding ratio of TPY-TTF LMWG with $Zn(NO_3)_2$ in CPG (i.e. 1:2). Therefore, unit formula weight of Zn-TPY-TTF CPG is 2212.94. Therefore, 1 mg Zn-TPY-TTF CPG catalyst contains 0.451 μmol and the corresponding TON was found to be 23.5 in 20 h. Similarly, TON was also calculated for CO formation in presence of Zn-TPY-TTF CPG photocatalyst. The detailed TON calculation for $H_2$ production and $CH_4$ formation in presence of Pt@Zn-TPY-TTF CPG are provided in Supplementary Table 3.

*Apparent quantum efficiency.* The apparent quantum efficiency (AQE) is defined by the ratio of the effective electron used for product formation to the total input photon flux. The AQE is calculated by using the below formula.

$$\text{AQE(\%)} = \left[\frac{\text{Effective electrons}}{\text{Total photons}}\right] \times 100\% = \left[\frac{n \times Y \times N}{\theta \times T \times S}\right] \times 100\% \tag{3}$$

Where, n is the number of electrons used in the photocatalysis process, $Y$ is the yield of evolved gas from the sample (mol), $N$ is the Avogadro's number ($6.022 \times 10^{23}$ mol$^{-1}$), $\theta$ is the photon flux, $T$ is the irradiation time, and $S$ is the illumination area. The photon flux was calculated at 400 ± 10 nm, 450 ± 10 nm, 500 ± 10 nm, 550 ± 10 nm, 600 ± 10 nm, 650 ± 10 nm, and 700 ± 10 nm by using separate bandpass filters with the help of power meter. The power meter (model: LaserCheck: 0623G19R) used for the experiment was purchased from Coherent.

Since the reduction of a water molecule to the $H_2$ molecule requires two-electron ($n = 2$). The following calculation is based on data from water photoreduction with Zn-TPY-TTF CPG for 1 h, AQE was calculated as 0.76% at 550 ± 10 nm (the amount of $H_2$ evolved in 1 h was found to be 0.094 μmol). Meanwhile, the AQE of $H_2$ for the Pt@Zn-TPY-TTF CPG was also calculated at the various wavelength (400–700 nm), and the maximum was found at 550 ± 10 nm, which was calculated to be 14.47%. The detailed calculation for $H_2$ production in presence of Pt@Zn-TPY-TTF CPG at different wavelength is provided in Supplementary Table 5.

The reduction of a $CO_2$ molecule to the CO molecule requires two-electron ($n = 2$). The following calculation is based on data from $CO_2$ photoreduction with Zn-TPY-TTF CPG at 550 ± 10 nm, AQE was calculated as 0.96%. (the amount of CO evolved was found to be 0.121 μmol in 1 h). Similarly, the reduction of a $CO_2$ molecule to the $CH_4$ molecule requires eight-electron ($n = 8$). Therefore, the AQE for $CH_4$ formation by Pt@Zn-TTF-TPY CPG was calculated at various wavelengths (400–700 nm) and found the maximum efficiency at 550 ± 10 nm as 0.81%. The detailed calculation for $CO_2$ photoreduction to $CH_4$ in presence of Pt@Zn-TPY-TTF CPG at different wavelength is provided in Supplementary Table 9.

## Data availability

All other data supporting the findings are available in the article as well as the supplementary information files and can be found from the authors on reasonable request.

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

## Acknowledgements

P.V., F.A.R. and P.S. acknowledge the Council of Scientific and Industrial Research (CSIR), Government of India, for the Senior Research Fellowship (SRF). S.K.P. thanks Science and Engineering Research Board (SERB), Dept. of Science and Technology (DST), and J. C. Bose National Fellowship, Government of India, for funding (Project No. 4554). T.K.M. acknowledge SERB, DST, Govt. of India for financial support (Project no. CRG/2019/005951).

## Author contributions

P.V. and T.K.M. designed research. P.V. performed the experimental works and analysed data. P.V., A.S. and T.K.M. contributed to analysing experimental data and wrote the manuscript. F.A.R., P.S. and S.K.P. carried out the computational studies. S.N. performed the transient absorption studies. All the authors discussed the results and commented on the manuscript.

## Competing interests

The authors declare no competing interests.
