## [Peer review file · Nature Communications]

REVIEWER COMMENTS

Reviewer #1 (Remarks to the Author):

This is an interesting paper where the authors describe a gelator that can be used as part of a photocatalytic system to make H₂ and also reduce CO₂. There are a lots of interesting data here, but also lots of questions.

The authors seem to have missed a number of key papers on hydrogen evolution using low molecular weight gelators. I suggest the authors carry out a thorough literature search and update their reference list accordingly. There are also many key papers describing the use of small organic molecules as part of a photocatalytic system that are relevant here and I suggest that the authors include some of these. Many of those I would expect to see, and I think are important to the main paper, are in the Supporting Information, so it is clear the authors are aware of them.

The inversion test absolutely does not prove gelation. It simply shows that flow does not occur over the timescale of the experiment. This sentence should be revised. Rheology is needed to show gelation has occurred.

Drying such gels is very likely to lead to morphological changes (especially when such mixed solvent systems with different volatility are used). I am always suspicious when flake like morphologies are described in such systems as these often arise from drying issues. If the authors do not have any evidence that there are no drying artefacts, a sentence stating that there may be issues needs to be included. This is also true for the pXRD, where the data is for the xerogel and so may well not represent what is happening in the solvated gel phase. This of course may not matter in the scheme of things since the authors mainly use the xerogel for the H₂ study. However, this raises a question about the materials design – is the gel state at all important? The data in Figure S16 seem to imply that the gel and xerogel have the same rate of reaction which seems surprising as I would have expected there to be a higher available surface area for reaction in the xerogel case, or at least an induction period where the gel liquid phase (which contains non-aqueous solvents) equilibrates with the bulk.

Further morphological questions arise from the H₂ evolution experiments. The xerogels are added to a water/TEA mixture. Are there any changes in morphology when this happens or any dissolution? What are the solubilities of the different components in this mixture? What happens if the components of the gel are simply suspended in such a mixture and stirred without the pre-formation of the xerogel? Also (for later experiments where ACN/H₂O mixtures of different composition are used), what is the solubility in ACN?

For these experiments, is TEA necessary or can another sacrificial donor be used?

Are there temperature effects on irradiation?

Is the same activity seen if the components are simply mixed? Is the morphology important? If (for example) different drying conditions are used, can the morphology be affected and does this then lead to different activity?

Recent papers suggest that a fibrous structure is beneficial for H₂ evolution and should be included.

For TPY-TTF, the NMR in Fig 3 (ESI) does not seem to have the integrations that the list of NMR peaks suggest. For example, peak d and f integrate to 8 H each (as shown in the list), but peak g integrates to 3H, but is attributed to 4H in the list. The list of data should be the number of protons integrated, not simply the number expected. Why is there this discrepancy?

For the gelation, the method reads as heated to for a viscous solution and then this gels on standing. How were these conditions determined? What are the structures in the viscous solution and are these the same each time? How reproducible is this approach?

For the data in Fig S39, the shape of the methane seems to suggest that the rate of methane production is increasing with time before degassing in each case and looks like the inverse shape of the CO evolution curve in (a) which seems to show the rate decreasing with time before the purge. Why is this?

As a minor point, the clouds over the sun symbol on each day for Fig 4f, Fig 5e, Fig. 6c and 6f is not clear to me. Is this meant to represent the weather that day? It would be more useful to provide some more detailed information such as overall temperature, number of hours of sunlight etc. These symbols are used across the world, but I suspect that they mean different things depending on one's location!

In Fig. S16, was the stability test carried out for the gel or xerogel? This needs to be clarified in the caption.

Reviewer #2 (Remarks to the Author):

Pending (as explained earlier, we will send you this report as soon as possible after this reviewer provides some more information)

Reviewer #3 (Remarks to the Author):

The manuscript entitled as "Charge-Transfer Regulated Visible Light Driven Photocatalytic H₂ Production and CO₂ Reduction in Tetrathiafulvalene Based Coordination Polymer Gel" reported the synthesis of a soft processable metal-organic hybrid semiconducting material and studied the photocatalytic activity towards H₂ production and CO₂ reduction. Through introducing Pt NPs, the H₂ evolution was much enhanced and the product in CO₂ reduction changed from CO to CH₄. In the manuscript, some of the conclusions put forward by the authors did not provide favorable evidence to confirm, some of the conclusions is contradictory, and even some opinions are inappropriate. Therefore, it is not recommended to accept the manuscript for publication in Nature Communications in current state.

1) Pt nanoparticles are famous hydrogen production catalysts in solution. From the TEM provided by the author, we can see that there are many small spherical particles scattered around the nanowires. Are these Pt nanoparticles? Moreover, many published works have proved that during the catalysis process, Pt nanoparticles will fall off from the structure and be free in the solution for catalysis. The author did not take this into consideration. In addition, the possible loading positions of Pt on the structure, or the interaction force between Pt nanoparticles and structures are not clear.

2) The manuscript emphasizes the influence of charge transfer route on catalysis while the authors pointed out that the catalytic effect is also related to other factors, such as morphology. But the author compares the activity of catalysts with different morphology. These results cannot support the authors' conclusion.

3) Regarding the electron transfer path after loading Pt, the author claims that electrons are transferred from Zn-TPY-TTF to Pt. In view that Pt nanoparticle can produce surface plasmon resonance under illumination, which promotes the absorption of visible light, the current characterization cannot directly prove the authors' point of view. It is recommended that the author conduct ultra-fast tests.

4) The authors compared the energies of the LUMO for TTF(CONH₂)₄ and [Zn(terpy)₂] to illuminate that excited-state electron transfer is energetically favourable from TTF(CONH₂)₄ core to [Zn(terpy)₂] centre. This is inappropriate to one molecule. This viewpoint will mislead readers.

5) For the stability test, the author made a morphological characterization, which is not enough to

explain the stability of the material. More characterizations are needed, such as elemental analysis, PXRD, etc after reaction.

6) Recycling experiment means that the catalyst can be retrieved and reused. However, the author's operation is to blow gas into the system again, which is not a true reuse.

7) The author conjectured the transformation path of the catalyst in the CO₂ reduction reaction. It is suggested that the author perform in-situ characterization of DRIFT to illuminate this.

8) As we all know, in the CO₂ reaction, due to the lower potential of hydrogen production, hydrogen reduction is a competitive reaction of CO₂ reduction. The Zn-TPY-TTF is an excellent hydrogen reduction catalyst. Why in the presence of water in the CO₂ reduction system, hydrogen production can be effectively inhibited, please explain.

9) The content of the article is a bit lengthy, some of the characterization of the comparison and the related the catalytic results are recommended to be moved to the supporting information.

Reviewer #4 (Remarks to the Author):

In this manuscript, the authors report the photocatalytic performance of a CT coordination polymer gel Zn-TPY-TTF for both H₂ production and CO₂ reduction under both visible light and sun light illumination. This work is interesting and the authors have performed the characterization and catalytic measurements thoroughly. I would recommend the publication of this work in nature communication if the authors can address one minor concern.

1) More details of lifetime measurements are required to understand the charge transfer process. 510 nm was used to excite the samples. Which species is excited under 510 nm excitation? Can the authors selectively excite one of the component? In addition, what is the emission wavelength that TRPL was collected? What does the emission collected for TRPL represent for? The answers to these questions are important given that intramolecular and intermolecular charge transfer are the key to the catalytic performance of the catalyst in this work.

**JAWAHARLAL NEHRU CENTRE FOR ADVANCED SCIENTIFIC
RESEARCH**

Jakkur, Bangalore - 560 064, India

Nature Communication

Reviewer's response (Manuscript No: NCOMMS-20-42521)

We would like to thank all the reviewers for their insightful comments and suggestions, which have helped us to improve the quality of the manuscript. Following the suggestions, we have thoroughly revised the manuscript with additional experimental results and highlighted those changes in yellow. We believe that this revised version of the manuscript will substantiate our work and will be suitable for publication in Nature Communications.

Reviewer #1

(Remarks to the Author): This is an interesting paper where the authors describe a gelator that can be used as part of a photocatalytic system to make H₂ and also reduce CO₂. There are a lot of interesting data here, but also lots of questions.

Response: We are thankful to the reviewer for appreciating our work and constructive suggestions and comments. We are grateful to the reviewer for in-depth evaluation of our manuscript. We have addressed all the concerns raised by the reviewer and incorporated all the suggestions in the revised version of the manuscript.

1) The authors seem to have missed a number of key papers on hydrogen evolution using low molecular weight gelators. I suggest the authors carry out a thorough literature search and update their reference list accordingly. There are also many key papers describing the use of small organic molecules as part of a photocatalytic system that is relevant here and I suggest that the authors include some of these. Many of those I would expect to see, and I think are important to the main paper, are in the Supporting Information, so it is clear the authors are aware of them.

Answer: We thank the reviewer for the suggestion. We have added references related to the low molecular weight gelator-based photocatalytic systems in the main text. We hope that the manuscript now carried a balance citation for photocatalytic H₂ evolution and CO₂ reduction following the journal guideline.

2) The inversion test absolutely does not prove gelation. It simply shows that flow does not occur over the timescale of the experiment. This sentence should be revised. Rheology is needed to show gelation has occurred.

Answer: We thank the reviewer for suggesting the rheology experiment. We have performed strain-sweep rheology experiments for both, TPY-TTF OG and Zn-TPY-TTF CPG at 25° C. The value of storage modulus (G') and loss modulus (G'') in both, TPY-TTF OG and Zn-TPY-TTF CPG moved constantly in the linear viscoelastic (LVE) region, consisting of the larger G' value under less strain range as compared to the G'', indicating for the stable viscoelastic nature which is a characteristic for the gel formation. Moreover, value of the G'

and G'' moved constantly up to 0.01% of strain for the Zn-TPY-TTF CPG was found to be ~ 10 times larger than OG, illustrating higher stability in the former case which is possibly due to the coordination of the Zn^{2+} ion (Chem. Commun., 2007, 2802). The discussion and corresponding figures are added in the revised manuscript and supporting information (SI) as Supplementary Figure 8 (Figure R1) for TPY-TTF OG and Supplementary Figure 15 (Figure R2) for Zn-TPY-TTF CPG.

Figure R1: Strain amplitude sweep rheology plot for TPY-TTF OG at 25° C. The closed symbols and open symbols represent the storage modulus (G') and the loss modulus (G''), respectively.

Figure R2: Strain amplitude sweep rheology plot for Zn-TPY-TTF CPG at 25° C. The closed symbols and open symbols represent the storage modulus (G') and the loss modulus (G''), respectively.

3) Drying such gels is very likely to lead to morphological changes (especially when such mixed solvent systems with different volatility are used). I am always suspicious when flake like morphologies is described in such systems as these often arise from drying issues. If the authors do not have any evidence that there are no drying artefacts, a sentence stating that there may be issues needs to be included. This is also true for the PXRD, where the data is for the xerogel and so may well not represent what is happening in the solvated gel phase. This of course may not matter in the scheme of things since the authors mainly use the xerogel for the H_2 study. However, this raises a question about the materials design – is the gel state at all important?

Answer: We are thankful to the reviewer for this concern and agree with the comment that the mixed solvent systems may influence the morphology as well as PXRD pattern. In this regard, we have attempted to dry the gel under different conditions and subsequently, collected the images by FE-SEM (Figure R3) which was found similar irrespective to the drying condition.

Therefore, the fibrous morphology was likely to be an inherent feature of the CPG. Moreover, the AFM images of TPY-TTF OG show the stacked sheet-like morphology, whereas, in the similar solvents system, the Zn-TPY-TTF CPG shows distinct nano-ribbon-like morphology. This further confirms that the fibrous morphology of the CPG is the intrinsic property of the material, driven through metal coordination in a self-assembled 3D network rather than an artifact that can appear due to the drying issue. Further, we have also dried the gels (both OG and CPG) through critical point drying and we did not observe any change in morphology. Details are given in the response to comment of reviewer 2 (point 4).

Furthermore, we have recorded the PXRD pattern for the CPG in gel state as per the reviewer's suggestion which is found to be similar to the xerogel (**Figure R4**, added in revised SI, Supplementary Figure 20). Next, we would like to elaborate on the importance of gel for the photocatalytic activity of the material. As a control, we have prepared a coordination polymer (CP) of TTF-TPY with Zn^{2+} ion. Notably, the FE-SEM images of Zn-TTF-TPY CP showed aggregated spherical morphology, completely different than what we realized in case of Zn-TTF-TPY CPG (**Figure R5**, this experiment was described in first submission; SI section S7.2). It is noteworthy to mention that the fibrous nanomorphology is well documented for providing high surface area as compared to the aggregated structure and thus, CPG is expected to display better catalytic activity (*Nature Chem.*, 2014, 6, 964; *small*, 2014, 10, 1272; *Sci Rep.*, 2017, 7, 13401; *J. Am. Chem. Soc.* 2015, 137, 15241; *Environ. Sci. Technol.*, 2019, 53, 3, 1564). Indeed, the photocatalytic activity examined for both, CP and CPG, under similar conditions has confirmed that the later one showed three times better H_2 production, emphasizing the crucial role of the fibrous morphology in the catalysis which indeed obtained in the gel state. The fibrous morphology of the Zn-TPY-TTF CPG was found to be retained in both, gel and xerogel state and as a consequence, the photocatalytic performances of CPG were found to be similar in both gel and xerogel state. Nevertheless, we have performed the catalysis in the xerogel state because of easy handling.

Figure R3. FE-SEM images for Zn-TPY-TTF CPG dried under different conditions for 8 h (a) heating at 40°C under vacuum, (b) heating at 80°C under vacuum.

Figure R4: PXRD for Zn-TPY-TTF CPG in the gel state.

Figure R5: Comparison of morphology and corresponding photocatalytic activity for **Zn-TPY-TTF Polymer** and **Zn-TPY-TTF CPG**.

4) The data in Figure S16 seem to imply that the gel and xerogel have the same rate of reaction which seems surprising as I would have expected there to be a higher available surface area for reaction in the xerogel case, or at least an induction period where the gel liquid phase (which contains non-aqueous solvents) equilibrates with the bulk.

Answer: We thank the reviewer for raising this concern. As mentioned in the response to previous comment, the photocatalytic activity has a profound impact on the nano-structuring of the material and the fibrous morphology remains intact in both, gel and xerogel state, and therefore, showed similar photocatalytic H₂ evolution. Furthermore, we would like to mention that the Zn-TPY-TTF CPG in either gel or xerogel state, was first dispersed uniformly in aqueous medium (38 ml water and 2 ml TEA) by the sonication for both, HER and CO₂RR, and then utilized for photocatalysis. Thus, the induction period in the gel state was reached prior to photocatalysis measurements.

5) Further morphological questions arise from the H₂ evolution experiments. The xerogels are added to a water/TEA mixture. Are there any changes in morphology when this happens or any dissolution? What are the solubilities of the different components in this mixture? What happens if the components of the gel are simply suspended in such a mixture and stirred without the pre-formation of the xerogel?

Answer: We truly appreciate the reviewer for raising this concern. We have recorded FE-SEM and TEM images for the Zn-TPY-TTF CPG after the 6th cycle of the post-photocatalytic run and nanofibrous morphology was remained intact similar to the as-synthesized sample. This indicates the stability of the material in the course of photocatalysis. We also analyzed the photocatalytic solution through inductively coupled plasma optical emission spectroscopy (ICP-OES). Importantly, no zinc traces were observed in the ICP-OES analysis of the solution examined after photocatalytic cycles. The stability of the photocatalyst was also confirmed by PXRD and FT-IR analysis.

Next, the CPG was found to be insoluble in the mixture of 5 vol % TEA in water but formed good dispersion. The individual components of the Zn-TPY-TTF CPG are comprised of Zn salt, TPY-NH₂, and TTF-COOH. The Zn salt is soluble in water whereas the other two components, TPY-NH₂ and TTF-COOH, form fine dispersion in water but becomes soluble in the mixture of 5 vol % TEA in water (**Table R1**). Next, photocatalytic H₂ production was examined for individual components as well as blending them under the similar condition as employed for the Zn-TPY-TTF CPG. Notably, maximum H₂ production realized by the

physical mixing of the individual components of the catalyst was calculated to be less than 0.2 mmol which is around forty-folds lesser than the obtained value for the Zn-TPY-TTF CPG (**Figure R6**, added in revised SI, Supplementary Figure 29). This signifies the importance of judiciously designed CPG towards realizing photocatalytic property.

Table R1. Solubilities of different components in water, acetonitrile, and mixture of these solvents with 5 vol % of TEA.

Components	Water	Acetonitrile (ACN)	Water with 5 vol % TEA	ACN+Water (3:1) with 5 vol % TEA
Zn(NO ₃) ₂	✓	✓	✓	✓
TPY-NH ₂	✗	✓	✓	✓
Zn(TPY) ₂ complex	✗	✗	✗	✗
TTF(COOH) ₄	✗	✗	✓	✓
TPY-TTF LMWG	✗	✗	✗	✗
TPY-TTF OG	✗	✗	✗	✗
Zn-TPY-TTF CPG	✗	✗	✗	✗
Pt@Zn-TPY-TTF CPG	✗	✗	✗	✗

Figure R6. Comparison of the photocatalytic hydrogen evolution upon blending the molecular components with the Zn-TPY-TTF CPG.

6) Also (for later experiments where ACN/H₂O mixtures of different compositions are used), what is the solubility in ACN?

Answer: We thank the reviewer for the concern. The Zn-TPY-TTF CPG is insoluble in all the common organic solvents including acetonitrile as well as the mixture of the ACN/H₂O. However, it forms a stable dispersion in both, ACN and ACN/H₂O solvent systems (**Table R1** as provided in response to the previous comment).

7) For these experiments, is TEA necessary or can another sacrificial donor be used?

Answer: We thank the reviewer for this suggestion. As per reviewer's suggestion, we have utilized different sacrificial electron donors for photocatalytic HER with Zn-TPY-TTF CPG. However, highest H₂ production was achieved using TEA as a sacrificial reagent (**Figure R7**, added in revised SI, Supplementary Figure 26).

Figure R7: Photocatalytic water reduction with Zn-TPY-TTF CPG xerogel using different sacrificial agents (0.35 mol L^{-1}).

8) Are there temperature effects on irradiation?

Answer: We are thankful to the reviewer for raising this concern. The possibility of the temperature effect on the hydrogen evolution due to the constant light irradiation on the reaction medium has been investigated by arranging the cold-water circulation around the reaction medium (**Figure R8**, added in revised SI, Supplementary Figure 23). Importantly, we found a maximum increase of 2°C in the absence of water circulation for the used light source, which is unlikely to change the photocatalytic activity. Furthermore, the photocatalytic H_2 evolution was found to be similar with or without water circulation around the reaction medium, confirming that the minimal rise in temperature as the consequence of constant irradiation does not affect the photocatalytic performance of the material. We have given the image of the photocatalytic setup used for this study in order to elucidate the comment.

Figure R8. Setup for controlled photocatalysis under laboratory condition; (1) Xenon lamp (Newport 300 W), (2) IR filter with cold water circulation, (3) Visible bandpass filter, (4) Cooling vessel, (5) Photocatalytic reaction cell, (6) Water circulation to maintain constant reaction temperature, (7) Magnetic stirrer.

9) Is the same activity seen if the components are simply mixed? Is morphology important? If (for example) different drying conditions are used, can the morphology be affected and does this then lead to different activity?

Answer: We are thankful to the reviewer for this concern. The physical blending of the structural units showed significantly lower photocatalytic activity as compared to the **Zn-TPY-TTF CPG (Figure R9)**. As mentioned above (comment 5), the fibrous morphology of the CPG plays a crucial role in photocatalysis (**Figure R9**, added in revised SI, Supplementary Figure 29). This is well supported by the poor photocatalytic performances of the coordination polymer which has aggregated spherical morphology (**Figure R5**). Next, we have employed different drying conditions to the gel to form xerogel of the Zn-TPY-TTF CPG (**Figure R3**). The FE-SEM images revealed that the fibrous morphology of the sample was retained and hence the photocatalytic activity also remained similar.

Figure R9. Photocatalytic activity for different molecular components compares to the **Zn-TPY-TTF CPG** for hydrogen evolution experiments under visible light for 6 h.

10) Recent papers suggest that a fibrous structure is beneficial for H₂ evolution and should be included.

Answer: We thank the reviewer for the suggestion. We have cited relevant references to signify the critical role of fibrous structure for H₂ evolution (*Nature Chem.*, 2014, 6, 964; *small*, 2014, 10, 1272; *Sci Rep.*, 2017, 7, 13401; *J. Am. Chem. Soc.* 2015, 137, 15241; *Environ. Sci. Technol.*, 2019, 53, 3, 1564).

11) For TPY-TTF, the NMR in Fig 3 (ESI) does not seem to have the integrations that the list of NMR peaks suggests. For example, peak d and f integrate to 8 H each (as shown in the list), but peak g integrates to 3H, but is attributed to 4H in the list. The list of data should be the number of protons integrated, not simply the number expected. Why is there this discrepancy?

Answer: We thank the reviewer for pointing out an important issue. We have repeated the ¹H-NMR for TPY-TTF LMWG at 600 MHz NMR instrument upon drying the sample under high vacuum at 120°C for 8 h. ¹H-NMR collected for the dried TPY-TTF LMWG sample has given in the revised supporting information (Supplementary Figure 3) which display similar number of protons and integration as expected theoretically. Specially, peak g was attributed to the 4 protons rather than 3 protons.

12) For the gelation, the method reads as heated to for a viscous solution and then this gel on standing. How were these conditions determined? What are the structures in the viscous solution and are these the same each time? How reproducible is this approach?

Answer: We appreciate the reviewer's point and would like to clarify the concern. The optimized condition for gelation of the TPY-TTF was achieved after several attempts made through permutation-combination of various solvents mixtures considering solubility of the TPY-TTF LMWG (**Table R2**, added in revised SI, Supplementary Table 1). It is noteworthy to mention that the TPY-TTF was readily soluble in the mixture of dichloromethane and methanol, however, insoluble in water. Therefore, playing with different solvent combinations as given in the table, we have optimized a solvent ratio of 2:1:1 of the methanol, dichloromethane and water, respectively, to get the desired gel material. We have repeated the gelation in the above given optimized condition at least 18-20 times and consistently obtained the gel material. Therefore, gelation under the optimized condition is highly reproducible and gel can be achieved easily. Gelation of the CPG (Zn-TPY-TTF) was also obtained under a similar condition as optimized for the OG. Next, formation of the CPG was reproduced 8-10 times and found to be equally effective as observed for the first time, confirming good reproducibility. Notably, the rheology experiment was performed for the three different batches of Zn-TPY-TTF CPG. This shows a similar linear viscoelastic (LVE) region with a fluctuation of $\pm 5\%$ (**Figure R10**). The rheology study provides experimental proof for the reproducibility of the gel formation. Our results demonstrated that the gelation in CPG proceeds from a clear solution through a turbid suspension and finally yielded the gel material. Therefore, we analyzed morphology of the turbid viscous solution prior to gel formation for Zn-TPY-TTF CPG as per the reviewer's suggestion which displayed self-assembled sponge-like network which eventually transforms to the interconnected fibrous morphology as time progress (**Figure R11**). This reflects that the gelator forms self-assembly towards the formation of an extended 3D network in presence of Zn^{2+} ion. We analysed the morphology for three trials for the viscous turbid solutions of gelator and found similar results, confirming the reproducibility of the pathway of the gel formation.

Table R2: Gelation ability of TPY-TTF LMWG in different conditions:

No	Solvents Composition	Solvent Ratio	Heating/Cooling	Gelation ability
1.	MeOH: DCM	(1:1)	60°/25° C	Solution
2.	MeOH: H ₂ O	(1:1)	60°/25° C	Precipitate
3.	DCM: H ₂ O	(1:1)	60°/25° C	Precipitate
4.	MeOH: DCM: H ₂ O	(1:1:1)	60°/25° C	Precipitate
5.	MeOH: DCM: H ₂ O	(2:2:1)	60°/25° C	Partial gel
6.	MeOH: DCM: H ₂ O	(2:1:1)	60°/25° C	Gel*

*CGC (Critical Gelator Concentration) = **0.005 mmol**; DCM= Dichloromethane; MeOH= Methanol; H₂O= Water.

Figure R10. Strain amplitude sweep for **Zn-TPY-TTF CPG**. (error bars represent the deviation in three measurements)

Figure R11. FE-SEM image for the viscous solution of Zn-TPY-TTF before gelation; (a) low resolution; (b) high resolution.

13) For the data in Fig S39, the shape of the methane seems to suggest that the rate of methane production is increasing with time before degassing in each case and looks like the inverse shape of the CO evolution curve in (a) which seems to show the rate decreasing with time before the purge. Why is this?

Answer: We are thankful to the reviewer for this concern. A similar signature for the photocatalytic methane as well as CO production as the function of time is reported earlier in the literature (*Chem. Rev.* 2019, **119**, 3962). However, we want to clarify that the production CO, as well as CH₄, has nothing related to the degassing process. It is noteworthy that the CO₂ reduction to CO and CH₄ are the two- and eight electron reduction processes, respectively, and therefore, has different reaction kinetics. Moreover, we have repeated the recyclability experiment as per one of the reviewer's suggestions (2nd reviewer's comment 10 and 3rd reviewer's comment 6). After each catalytic run, the catalyst has been recovered and again dispersed in the fresh solvent mixture to perform the photocatalysis for four addition cycles (Supplementary Figures 27, 39, 47 and 54). Importantly, the photocatalytic activity of the recovered sample towards CO and CH₄ formation was found to be retained > 99% and > 95%, respectively. We have explained the fact in the revised manuscript and added the details of experiments in the revised SI.

14) As a minor point, the clouds over the sun symbol on each day for Fig 4f, Fig 5e, Fig. 6c and 6f is not clear to me. Is this meant to represent the weather that day? It would be more useful to provide some more detailed information such as overall temperature, number of

hours of sunlight etc. These symbols are used across the world, but I suspect that they mean different things depending on one's location!

Answer: We are thankful to the reviewer for the suggestion. The cloud over the sun symbol was representing the weather on that day. Moreover, as per reviewer's suggestion, we have added temperature, as well as the number of hours of sunlight irradiation on that day in addition to the weather symbol in the revised manuscript (Fig 4g, Fig 5e, Fig. 6c, and Fig. 6f).

15) In Fig. S16, was the stability test carried out for the gel or xerogel? This needs to be clarified in the caption.

Answer: We are thankful to the reviewer for this suggestion. The photocatalytic stability was examined in the xerogel state of the catalyst. This information has been added in the caption of the figure in the revised SI (Supplementary Figure 27).

Reviewer 2:

Title: Charge-Transfer Regulated Visible Light Driven Photocatalytic H₂ Production and CO₂ Reduction in Tetrathiafulvalene Based Coordination Polymer Gel.

Based on comments below this reviewer is uncertain if the paper should be published in *Nature Communications*. It would require a major revision, new experiments, and clarification of points raised:

Response: We thank the reviewer for his/her constructive suggestions and comments. We found the reviewer's comments are extremely helpful to improve the overall quality of the manuscript. We have addressed all the concerns raised by the reviewer and the manuscript has been revised accordingly.

- 1) This manuscript reports on a low molecular weight gelator (LMWG), consisting of a tetrathiafulvalene (TTF) as a central aromatic core and terpyridine (TPY) side groups. The molecule is capable of forming an organogel, with or without the addition of zinc ions. The resulting hydrogel was reported to be photo-catalytically active for the production of hydrogen and CO₂ reduction. This work is potentially interesting to readers of *Nature Communications*, but in its current form, the manuscript has some major flaws that must be addressed before consideration for publication.

Answer: We thank the reviewer for appreciating our work and highlighting the work to be of general interest to the readers of *Nature Communications*. We have performed several experiments as well as theoretical calculations as suggested by the reviewer. The point-wise explanation to the reviewer's concerns is given below and the manuscript has been revised accordingly.

- 2) The authors should provide proof of purity for the LMWG. Based on the ¹H NMR characterization provided, it seems that the gelator molecule may not be pure. Specifically, the peak selection and integrals of the ¹H NMR spectra in Supplementary Figure 3 were not done properly. Given how broad the peaks are, even in DMSO, the authors should acquire the spectrum in a different solvent or heat it to a higher temperature. Furthermore, in the ¹³C NMR spectrum shown in Supplementary Figure 4, there are multiple peaks that have not been picked and reported. The synthetic protocol only reports 15 carbon resonances when 18 are expected. In any case, better evidence of purity should be given since a valid TON can only be reported for a pure molecule!!

Answer: We are thankful to the reviewer for the suggestion. We have dried TPY-TTF LMWG under vacuum at 120 °C for 12 h to ensure the removal of solvent molecules. Next, ¹H and ¹³C-NMR spectra were recorded and provided in the revised supplementary file (Supplementary Figures 3 and 4). Notably, peak positions and integrations of the ¹H and ¹³C-NMR spectra are in good agreement with the number of proton and carbon present in the molecular structure of the LMWG, confirming the formation of the desired compound. Further, well-matched molecular mass obtained by the HR-MS spectrum has provided strong support to the structure elucidation of the LMWG (Supplementary Figure 5).

- 3) A major point of the manuscript is that the intramolecular charge-transfer (CT) interaction (between TTF and TPY) is responsible for photocatalytic activity. However, based on the energy levels given by the authors, this looks improbable. From the main text (and Figure 3c), the authors claim that this would be a thermodynamically feasible (or favored) process,

but the LUMO of $[\text{Zn}(\text{TPY})_2]^{2+}$ lies at -1.92 eV, just under the level of TTF (-1.90 eV). A larger energy difference between those two levels is needed to account for the reorganization energy in water.

Answer: We are thankful to the reviewer for this concern. As per reviewer's comment, we have recalculated the band alignment by considering the active catalytic units. Thus, we have considered $\text{TTF}(\text{CONH}_2)_4$ and $[\text{Zn}(\text{TPY})_2]^{2+}$ as a donor and acceptor moiety, respectively, and excluded NO_3^- anion for optimization as the nitrate ion is just to neutralize the complex ($[\text{Zn}(\text{TPY})_2]^{2+}$) rather participating in the catalysis. Notably, the revised calculation showed a large energy difference between LUMO of donor (-1.91 eV) and acceptor (-2.17 eV) in aqueous medium, displaying good thermodynamic feasibility for the electron transfer (**Figure R12**). This revised calculation and figure have been added in manuscript (Fig. 3c).

Figure R12. HOMO-LUMO band alignments of $\text{TTF}(\text{CONH}_2)_4$ and $[\text{Zn}(\text{TPY})_2]^{2+}$ for thermodynamic feasibility of electron transfer in the aqueous medium.

4) The authors claim that the CT interaction is driven by intermolecular stacking of TTF and TPY and they propose a model. However, the analysis of the structure is confusing, and it is not at all clear how these two molecules would stack to form the planes separated by 3.7 \AA seen in TEM (Figure 1g) or the height of 7 nm observed in AFM (Figure 1e). The characterization of the TPY-TTF OG by SEM appears to give a stacked morphology which may not be representative of the morphology in the gel state. The methods section states that the xerogels were prepared for AFM, SEM, and TEM analysis by dispersion in ethanol. This is likely representative of the material as used in photocatalysis, but it is not representative of the material as a gel. Material preparation that better preserves the gel microstructure (ex. Critical Point Drying) may provide a better characterization of the gel state. It would also be helpful to know how the authors were able to exclude other models, such as the formation of TTF stacks. One of the references from the list (*Chem. Sci.*, 2011, 2, 2017) does indeed show an X-ray crystal structure of TTF molecules that stack on top of each other.

Answer: We thank the reviewer for this comment and suggestions. First, we want to clarify that the AFM image of the TPY-TTF OG in figure 1b shows stacked morphology which consists of several layered structures wherein height of the stacked-layer was neither calculated nor discussed in the manuscript. However, as per reviewer's concern, we have calculated distance between the layers in the stacked morphology based on AFM measurement (figure 1b) and found to be $3.4 \text{ nm} \pm 0.4 \text{ nm}$ (**figure R13**). Here, layered morphology as observed in the OG was formed by the assembly of TPY-TTF through intermolecular $\pi \dots \pi$ stacking which was confirmed by the TEM analysis and theoretical calculations (**Figure R29**, further discussed in response to the minor comment, point 2). The lattice fringes in the TEM images were found to be 3.7 \AA , suggesting the presence of π - π interactions (figure 1d: inset). These descriptions about the self-assembly in TPY-TTF OG have been clarified in the revised manuscript to avoid confusion. Similar to OG, the height of the nano-ribbon-like structure in the Zn-TPY-TTF CPG was found to be 7 nm as observed in AFM (figure 2f).

We have dried the sample upon heating ($80 \text{ }^\circ\text{C}$ under vacuum) as well as by the critical point drying (CPD) method (as suggested by the reviewer) and then we have characterized the morphology by SEM measurements. The morphology of the TPY-TTF OG after CPD was found to be similar as observed upon normal drying ($80 \text{ }^\circ\text{C}$ under vacuum). This confirmed that the morphology of the TPY-TTF OG in the gel state remained intact as in the xerogel state (**Figure R14**, added in revised SI as Supplementary Figure 11). Similarly, Zn-TPY-TTF was also dried through the CPD method and the morphology was found to be similar as observed upon normal drying (80°C under vacuum) as shown in **figure R15** (added in revised SI as Supplementary Figure 17). The detailed procedure of the CPD method is provided in the revised supporting information.

Next, we have optimized possible intermolecular packing models for Zn-TPY-TTF CPG using DFT calculations. We observed that the packing of TTF with $[\text{Zn}(\text{TPY})_2]^{2+}$ on top of each other was stabilized with a distance of 3.56 \AA which is in well agreement with the experimental observations obtained from PXRD and TEM analysis (**Figure R16**, added in revised manuscript as Fig. 2j and Supplementary Figure 22a). At the same time, another possibility of stacking through TTF...TTF unit on the top of each other was optimized which revealed the TTF...TTF distance $>11 \text{ \AA}$ due to steric repulsion among the $[\text{Zn}(\text{TPY})_2]^{2+}$ units attached to TTF core and therefore, ruled out the possibility of TTF---TTF stacking (**Figure R16b**). Furthermore, stacking of TTF with the terpyridine-based ligand is well documented in literature along with crystal structure as shown in **Figure R17** (*Eur. J. Inorg. Chem.* 2013, 6037; *Beilstein J. Org. Chem.* 2015, **11**, 1379; *Eur. J. Inorg. Chem.* 2014, 3912).

Figure R13. Morphological analysis for **TPY-TTF OG**; layered sheet-like morphology. (a) AFM image, (b) Height profile of the stacked sheet-like structure, (c) FESEM image, (d) HR-

TEM image (inset: showing lattice fringes). (e) Schematic representation of self-assembly of **TPY-TTF OG**.

Figure R14. SEM images of **TPY-TTF OG** after drying in different conditions: (a) under vacuum at 80°C and (b) under critical point drying (CPD).

Figure R15. SEM images of **Zn-TPY-TTF CPG** after drying in different conditions: (a) under vacuum at 80°C and (b) under critical point drying (CPD).

Figure R16. Optimized model for π - π stacking in **Zn-TPY-TTF CPG** system through DFT calculations. (a) Stacking of TTF-TPY unit. (b) Stacking of TTF-TTF units.

Figure R17. Crystal structure reported for stacking of TTF---TPY unit (*Eur. J. Inorg. Chem.* 2013, 6037; (a) The 50% thermal ellipsoid plot. Solvent molecules and anions are omitted for clarity and (b) Crystal-packing arrangement showing intermolecular C···C interactions).

5) The authors claim that they perform photocatalysis with their xerogel. However, the methods section states that the samples are prepared by sonicating 1 mg of xerogel in 38 mL of water “to make a homogeneous dispersion”, which would suggest that the photocatalysis was actually performed on a dispersion rather than a gel. If this is the case, then the material likely loses most of its 3D network structure. Furthermore, it is not clear if the material recovered after catalysis by centrifugation actually has a similar morphology to the original sample before sonication and dispersion (Figure S41). While the elemental composition from EDAX might be similar under the reported conditions, one would expect the supramolecular structure to be very different (such as smaller dimensions and more defects). Additional characterization of the material after catalysis, including XRD and UV-Vis, could give better evidence to this claim.

Answer: We thank the reviewer for this concern. We would like to clarify that the 1 mg of catalyst in the xerogel state was dispersed via sonication before performing the photocatalytic reaction. Whereas, photocatalytic studies in the gel state were performed by just dispersing the gel material in the catalytic medium. Importantly, the Zn-TPY-TTF CPG showed similar morphology in both, gel and xerogel state, indicating the robustness of the material which is a special feature of the Zn-TPY-TTF CPG. Furthermore, morphology of the material after post-catalytic run was examined by FESEM and TEM and found to be similar to the as-synthesized material, again confirming the robustness of the material (for Zn-TPY-TTF CPG - Supplementary Figure 55; **Figure R18** and Pt@Zn-TPY-TTF CPG - Supplementary Figure 59; **Figure R19**). The analysis showed that 3D interconnected nanofibrous morphology remains intact. Additionally, as per the reviewer’s suggestions, we have performed FT-IR, PXRD and UV-Vis absorption studies after collecting the sample from post-catalytic run (for Zn-TPY-TTF CPG; Supplementary Figure 55-58; for Pt@Zn-TPY-TTF CPG; Supplementary Figure 59-62). The PXRD pattern and absorption spectrum of the recollected sample after the catalytic run was found similar to the as-synthesized material, supporting that the integrity of the material remained intact after the catalysis.

REVIEWER COMMENTS

Reviewer #1 (Remarks to the Author):

The authors have revised the manuscript significantly, adding much more detail and explanation. I was also asked to look into referee #2's comments. Overall, I think that the response addresses both my and referee 2's comments effectively.

As a couple of minor comments:

What frequency was the strain sweep carried out at for the rheology? It would also be best to provide some more details as to how the gel was prepared for rheology and how it was loaded on to the rheometer to ensure that no damage occurred on loading. In my experience, this is often one of the hardest aspects to ensure that the data are valid.

It would perhaps be useful to include a comment as to why the authors think that TEA gives the highest H₂ production out of all the sacrificial reagents. This is also seen for other systems and so is worth a note I think.

Reviewer #3 (Remarks to the Author):

The work entitled "Charge-Transfer Regulated Visible Light Driven Photocatalytic H₂ Production and CO₂ Reduction in Tetrathiafulvalene Based Coordination Polymer Gel" reports low molecular weight gelator (LMWG) constructed from Zn ions and TPY-TTF ligands, and Pt doped composites for photochemical CO₂ reduction and H₂ production. After revision, the quality of this manuscript is improved but some issues still need to be resolved.

- 1) The authors claim Charge-Transfer Regulated photochemical reaction while the related discussion about catalytic mechanism of Pt doped catalyst is insufficient.
- 2) Ultrafast absorption spectroscopy should be tested to illuminated the charge separation of Pt@Zn-TPY-TTF CPG.
- 3) When calculating the TON, the author uses the amount of Pt as the benchmark, indicating Pt is catalytic center. Therefore, the catalytic mechanism of Pt doped catalyst is different from that without Pt. However, there is not much discussion about the mechanism.
- 4) The binding sites of Pt obtained by theoretical calculations need to be further confirmed. The Pt model selected in the theoretical calculation is a single atom, while the Pt in the experiment is a nanocluster. It is suggested that the authors use synchrotron radiation to further explore the role of Pt. The clear structure is helpful to the study of catalytic mechanism.
- 5) The reason for the high CO selectivity under 75% conditions must be provided.
- 6) The amount of Zn should be further determined by ICP-MS.

Nature Communication

Reviewer's response (Manuscript No: NCOMMS-20-42521A)

Reviewer #1

(Remarks to the Author): The authors have revised the manuscript significantly, adding much more detail and explanation. I was also asked to look into referee #2's comments. Overall, I think that the response addresses both my and referee 2's comments effectively.

Response: We are thankful to the reviewer for appreciating our effort. We are grateful to the reviewer for the in-depth evaluation of our manuscript.

As a couple of minor comments:

1. What frequency was the strain sweep carried out at for the rheology? It would also be best to provide some more details as to how the gel was prepared for rheology and how it was loaded on to the rheometer to ensure that no damage occurred on loading. In my experience, this is often one of the hardest aspects to ensure that the data are valid.

Response: Rheological tests of gels were carried out using dynamic strain sweep tests at a constant frequency ($\omega = 1$ Hz). For each rheology measurement, gel was prepared in 10 ml glass vial. Next, approximately 20 mg of gel sample was loaded onto the rheometer plate with the help of a spatula in a single shot to avoid any damage to the loaded sample. Further, data accuracy was ensured by repeating these experiments a minimum of three times. We have added this discussion in the revised supporting information.

2. It would perhaps be useful to include a comment as to why the authors think that TEA gives the highest H₂ production out of all the sacrificial reagents. This is also seen for other systems and so is worth a note, I think.

Response: We thank the reviewer for the suggestion. We agree with the reviewer that in many reported studies, the use of TEA as a sacrificial reagent significantly enhances photocatalytic H₂ production as compared to other sacrificial reagents (*Energy Environ. Sci.*, 2020, 13, 1843; *J. Am. Chem. Soc.*, 2019, 141, 9063). Likewise, we also observed the highest H₂ production using TEA. The better performance of TEA as compared to other sacrificial reagents could be attributed to the high adsorption ability of TEA on the catalytic surface (*ACS Omega* 2019, 4, 11135), which eventually reduces the Schottky barrier and, as a consequence, facilitate the electron transfer (*ACS Appl. Mater. Interfaces.*, 2020, 12, 46267; *J. Am. Chem. Soc.*, 1988, 110, 4914).

Reviewer #3

(Remarks to the Author): The work entitled “Charge-Transfer Regulated Visible Light Driven Photocatalytic H₂ Production and CO₂ Reduction in Tetrathiafulvalene Based Coordination Polymer Gel” reports low molecular weight gelator (LMWG) constructed from Zn ions and TPY-TTF ligands, and Pt doped composites for photochemical CO₂ reduction and H₂ production. After revision, the quality of this manuscript is improved but some issues still need to be resolved.

Response: We are thankful to the reviewer for appreciating our work and constructive suggestions. We have addressed all the concerns raised by the reviewer and incorporated all the additional details in the revised version of the manuscript.

1. The authors claim Charge-Transfer Regulated photochemical reaction while the related discussion about catalytic mechanism of Pt doped catalyst is insufficient.

Response: We thank the reviewer for the suggestion. The step-wise catalytic mechanism for platinum nanoparticle doped CPG sample (Pt@Zn-TPY-TTF CPG) has been computed in detail in line with experimental results obtained from in situ DRIFT studies (Fig. R1; added in the revised manuscript as Fig. 7d, Supplementary Fig. 66-68). The results showed that the Pt nanoparticles play a pivotal role during the photocatalytic CO₂ reduction to CH₄. The plausible mechanism of CH₄ formation and corresponding discussion has been added in the revised manuscript and SI.

Fig. R1. Plausible mechanism computed for the CO₂ reduction to CH₄ by Pt@Zn-TPY-TTF CPG.

2. Ultrafast absorption spectroscopy should be tested to illuminated the charge separation of Pt@Zn-TPY-TTF CPG.

Response: We thank the reviewer for the suggestion. Ultrafast transient absorption spectroscopy was performed to check the possibility of electron transfer from Zn-TPY moiety to Pt nanoparticles in Pt@Zn-TPY-TTF CPG (Fig. R2; added in revised SI as Supplementary Fig. 42). The transient decay of Zn-TPY anion, which is formed due to electron transfer from TTF to Zn-TPY, is significantly faster in Pt@Zn-TPY-TTF CPG than Zn-TPY-TTF CPG (Fig. R2b & c). This additional transient absorption data support our proposal that C-T driven electron transfer from TTF to Zn-TPY to Pt nanoparticles that facilitate efficient catalytic activities in H₂ production and CO₂ reduction to CH₄ by Pt@Zn-TPY-TTF CPG. This discussion has also been added to the revised manuscript.

However, there are several recent reports in the area of photocatalysis in which the mechanistic aspects have been only explained mainly through TRPL study (*ACS Energy Lett.* 2020, 5, 669; *Adv. Funct. Mater.* 2019, 29, 1902992; *Angew. Chem. Int. Ed.* 2020, 59, 16902; *Angew. Chem. Int. Ed.* 2017, 56, 7876; *Chem. Commun.*, 2020, 56, 527; *Small* 2017, 13, 1603301; *Nature Commun.* 2019, 10, 4421; *Adv. Funct. Mater.* 2019, 29, 1808156).

Fig. R2. (a) Transient absorption spectra of Zn-TPY-TTF CPG dispersed in methanol at different time delays (1-100ps). Transient decay traces at 580 nm for Zn-TPY-TTF CPG (b) and Pt@Zn-TPY-TTF CPG (c). Solid lines in panels (b) and (c) are fitted curves.

3. When calculating the TON, the author uses the amount of Pt as the benchmark, indicating Pt is catalytic center. Therefore, the catalytic mechanism of Pt doped catalyst is different from that without Pt. However, there is not much discussion about the mechanism.

Response: We thank the reviewer for this comment. We agree with the reviewer that the photocatalytic mechanism for with and without Pt will be different. Therefore, a detailed catalytic mechanism for Pt@Zn-TPY-TTF CPG has been computed with the agreement of experimental results (in situ IR), which is provided in comment 1 and added to the revised manuscript as well as in supporting information.

4. The binding sites of Pt obtained by theoretical calculations need to be further confirmed. The Pt model selected in the theoretical calculation is a single atom, while the Pt in the experiment is a nanocluster. It is suggested that the authors use synchrotron radiation to further explore the role of Pt. The clear structure is helpful to the study of catalytic mechanism.

Response: We thank the reviewer for the suggestion. We agree with the reviewer that the binding site for the platinum could be elucidated using Pt nanocluster instead of a single atom of Pt. In order to get a clear idea, we have performed theoretical studies by considering Pt(111) nanoclusters of different sizes ($n=3, 4$) and elucidated binding sites at different positions. However, all our attempts of moving towards larger Pt nanoclusters increase the computational cost significantly. Importantly, the most significant observations that come out from the computed results are that the preferred binding site of the platinum remains independent of the cluster size (see Fig. R3), and the most suitable position for stabilization of Pt nanocluster irrespective of the cluster size is found to be similar as observed for single Pt atom (i.e., below the horizontal terpyridine units connected to Zn) as shown in Fig. R3 (added in revised SI as Supplementary Fig. 44).

We would like to mention that ICP, HR-TEM and PXRD analysis confirmed the presence, size and the exposed surface of the Pt nanoparticles in the form of Pt(111). It is not clear to us what type of additional information can be provided by the synchrotron experiments. Furthermore, there are several recent reports on Pt loaded photocatalyst materials where details have been provided without synchrotron study (*Nature Commun.*, 2018, 9, 1252; *Sci. Reports*, 2018, 8, 16198; *J. Mater. Chem. A*, 2014, 2, 692; *Angew. Chem. Int. Ed.*, 2013, 52, 5776; *Green Chem.*, 2010, 12, 212; *Chem. Commun.*, 2016, 52, 35; *Catal. Sci. Technol.*, 2020, 10, 5048; *ACS Catal.*, 2014, 4, 3644; *J. Am. Chem. Soc.*, 2012, 134, 11276; *J. Am. Chem. Soc.*, 2017, 139, 4789; *ACS Catal.*, 2017, 7, 8228; *J. Phys. Chem. C*, 2013, 117, 26415). In this manuscript, we have characterized Pt loaded samples (Pt@Zn-TPY-TTF CPG) similarly to the earlier reports.

Fig. R3. (a) Schematic of the structure of TTF and $[\text{Zn}(\text{TPY})_2]^{2+}$ stack system for possible loading positions of Pt nanoparticles. (b) Stabilization energies of Pt in kcal/mol corresponding to loading positions. The most suitable loading position was observed in close proximity to 4th 'N' atom of the central pyridine ring of the horizontal terpyridine unit. (c) Different optimized models were obtained from DFT calculations to illustrate the possible loading position of different Pt models on TTF and $[\text{Zn}(\text{TPY})_2]^{2+}$ stack in **Zn-TPY-TTF CPG** system and corresponding stabilization energies in kcal/mol.

5. The reason for the high CO selectivity under 75% conditions must be provided.

Response: We thank the reviewer for the suggestion. Importantly, CO₂ reduction to CO is a proton-coupled reduction process. Thus, the individual role of both the solvents, acetonitrile and water, as well as composition, play a very crucial role towards attaining high CO production in the course of photocatalytic CO₂ reduction. In brief, acetonitrile provides high solubility to the CO₂, whereas water acted as a proton source (*J. Electrochem. Soc.*, 2000, 147, 4164; *Angew. Chem. Int. Ed.*, 2019, 58, 12180; *J. Am. Chem. Soc.*, 2020, 142, 6188). Screening of the solvent mixture indicated that a combination of 75% acetonitrile and 25% water led to an optimized composition wherein maximum solubility of the CO₂ along with sufficient proton source can be achieved, and as a result, the highest CO production was realized. The explanation has been added in the revised manuscript.

6. The amount of Zn should be further determined by ICP-MS.

Response: We thank the reviewer for the suggestion. We have performed ICP analysis for as-synthesized Zn-TPY-TTF CPG sample as well as a recovered sample after photocatalysis. In both cases, the amount of Zn content was calculated to be 5.9 (± 0.2) wt%. We have added the relevant discussion at the appropriate place in the revised manuscript.

Finally, we are thankful to all the reviewers for their comments and constructive suggestions, which have helped to improve the quality of the manuscript significantly.

REVIEWER COMMENTS

Reviewer #3 (Remarks to the Author):

The authors have revised the manuscript significantly, most of the concerns have been addressed, except for the clear binding sites of Pt nanoparticles. X-ray absorption fine structure (XAFS) and X-ray absorption spectroscopy (XAS) based on synchrotron radiation source will be help for studying the local atomic and electronic structure of materials.

Nature Communication

Editor's response (Manuscript No: NCOMMS-20-42521B)

Comments of Editor:

As you can from the reports below, reviewer #3 believes that experimental evidence for Pt binding environments and local electronic structure are currently lacking without XAS. Reviewer #3 believes that this challenges the DFT model and the discussion on the charge-transfer mechanism for Pt incorporated material. After carefully evaluating the reviewer's concerns, while our editorial team does not request the XAS to be presented in your manuscript, we believe that sufficient caveats and limits of the DFT results must be added to resolve the reviewer's concern. In this regard, we believe that the detailed DFT discussion on Pt incorporated material (such as Line 358-367 and Line 487-522, and Figure 7c, d) should be moved to the supplementary information. Instead, a brief conclusion of DFT results with softened claim should be presented in the main manuscript. When discussing DFT results, the limits of the DFT model (and results) without unambiguous experimental confirmation of the Pt binding sites and electronic structure should be fully discussed in the main manuscript. Potential future effort and direction that should be taken to resolve this issue should also be discussed. Your revision should address the above editorial request before we can proceed further.

Response: We are thankful to the editor(s) for constructive suggestions and agreeing for not performing XAS. As per editor's suggestion, we have shortened the DFT results and softened the claim in the revised main manuscript. More importantly, the detailed discussion given in Line 358-367 and Line 487-522, and Figure 7c, d have been shifted to the SI.

Reviewer's response (Manuscript No: NCOMMS-20-42521B)

Comment of Reviewer 3:

The authors have revised the manuscript significantly, most of the concerns have been addressed, except for the clear binding sites of Pt nanoparticles. X-ray absorption fine structure (XAFS) and X-ray absorption spectroscopy (XAS) based on synchrotron radiation source will be helpful for studying the local atomic and electronic structure of materials.

Response: We thank the reviewer for appreciating and agreeing that most of his/her concerns have been addressed in the revision of the manuscript. Furthermore, we sincerely take the reviewer's suggestion of XAFS and XAS study for future reference and will try to execute while performing in-depth mechanistic studies of the photocatalysis.